# e-GAI: e-value-based Generalized $\alpha$-Investing for Online False Discovery Rate Control

**Yifan Zhang** [1]  **Zijian Wei** [1]  **Haojie Ren** [1]  **Changliang Zou** [2]

## Abstract

Online multiple hypothesis testing has attracted a lot of attention in many applications, e.g., anomaly status detection and stock market price monitoring. The state-of-the-art generalized $\alpha$-investing (GAI) algorithms can control online false discovery rate (FDR) on p-values only under specific dependence structures, a situation that rarely occurs in practice. The e-LOND algorithm (Xu & Ramdas, 2024) utilizes e-values to achieve online FDR control under arbitrary dependence but suffers from a significant loss in power as testing levels are derived from pre-specified descent sequences. To address these limitations, we propose a novel framework on valid e-values named e-GAI. The proposed e-GAI can ensure provable online FDR control under more general dependency conditions while improving the power by dynamically allocating the testing levels. These testing levels are updated not only by relying on both the number of previous rejections and the prior costs, but also, differing from the GAI framework, by assigning less $\alpha$-wealth for each rejection from a risk aversion perspective. Within the e-GAI framework, we introduce two new online FDR procedures, e-LORD and e-SAFFRON, and provide strategies for the long-term performance to address the issue of $\alpha$-death, a common phenomenon within the GAI framework. Furthermore, we demonstrate that e-GAI can be generalized to conditionally super-uniform p-values. Both simulated and real data experiments demonstrate the advantages of both e-LORD and e-SAFFRON in FDR control and power.

## 1. Introduction

The online multiple hypothesis testing problem arises from a range of applications. For example, regulators record the number of NYC taxi passengers every 30 minutes, aiming to detect the anomalous intervals corresponding to special events (Lavin & Ahmad, 2015); economists build online monitoring procedures based on monthly stock market prices to identify bubbles in financial series (Genoni et al., 2023); industrial factories monitor machine operation status in real time for fault detection, thereby enabling early warnings for potential system issues (Ahmad et al., 2017). These different scenarios can all be formulated as the online multiple testing problem, which is concerned with the investigation of an online sequence of null hypotheses. At each time $t$, we must immediately make a real-time decision on whether to reject the current hypothesis based on all the observed data so far, without having knowledge of future data or the total number of hypotheses.

Consider an online sequence of null hypotheses $\mathbb{H}_1, \ldots, \mathbb{H}_t, \ldots$. Define $\theta_t = 0/1$ if $\mathbb{H}_t$ is true/false for each time $t$ and a class of online decision rules $\boldsymbol{\delta}_t = \{\delta_j : j = 1, \ldots, t\}$, where $\delta_t = 1$ indicates that $\mathbb{H}_t$ is rejected and $\delta_t = 0$ otherwise. It is necessary to control the error rates of those decisions $\boldsymbol{\delta}_t$. A natural quantity to control is the false discovery rate (FDR) at target level $\alpha$ as introduced by Benjamini & Hochberg (1995), that is, the ratio of falsely rejected nulls to the total number of rejections. The online FDR is defined as:

$$\mathrm{FDR}(t) = \mathbb{E}[\mathrm{FDP}(t)] = \mathbb{E}\left[\frac{\sum_{j \in \mathcal{H}_0(t)} \delta_j}{\left(\sum_{j=1}^t \delta_j\right) \vee 1}\right] \leq \alpha,$$

where $\mathcal{H}_0(t) = \{j \leq t : \theta_j = 0\}$ is the true null set up to time $t$.

**Related works.** Methods for online FDR control were pioneered by Foster & Stine (2008), who proposed the so-called $\alpha$-investing (AI) strategy. It was further extended by the generalized $\alpha$-investing (GAI) procedure, which has served as the fundamental framework for online testing problem (Aharoni & Rosset, 2014; Javanmard & Montanari, 2018). GAI deals with a sequence of p-values $p_1, \ldots, p_t, \ldots$ and

[1]School of Mathematical Sciences, Shanghai Jiao Tong University, Shanghai, China [2]School of Statistics and Data Sciences, LPMC, KLMDASR and LEBPS, Nankai University, Tianjin, China. Correspondence to: Haojie Ren <haojieren@sjtu.edu.cn>.

*Proceedings of the 42$^{nd}$ International Conference on Machine Learning*, Vancouver, Canada. PMLR 267, 2025. Copyright 2025 by the author(s).

encompasses a wide class of algorithms to assign the testing levels $\alpha_1, \ldots, \alpha_t$ in an online fashion, effectively rejecting the $t$-th null hypothesis whenever $p_t \leq \alpha_t$ (Ramdas et al., 2017; 2018; Tian & Ramdas, 2019). The GAI ensures provable online FDR control only for the p-values with independence or positive regression dependence on a subset (PRDS; Benjamini & Yekutieli, 2001), which is rarely the case in practice; see definition of PRDS in Appendix B.1. In fact, many applications involve unknown complex dependence, such as time dependence in stock market prices. Hence, it is important to develop online multiple testing procedures under more general dependency conditions.

Xu & Ramdas (2022) proposed the SupLORD algorithm with the aim of false discovery exceedance control and discovered that, under a weaker baseline assumption, i.e., the null p-values are conditionally super-uniform as formalized in (1) in the subsequent text, it ensures valid FDR control at arbitrary stopping times. However, the SupLORD algorithm necessitates the selection of multiple parameters, which are intricately linked to its performance, and currently lacks well-established criteria for their optimal selection.

To deal with the problem introduced by dependence, another strategic direction is proposed by Wang & Ramdas (2022) to utilize e-values as potential alternatives to p-values as measures of uncertainty, significance, and evidence. The e-values have gained considerable attention, and many works have devoted significant effort to constructing valid e-values (Vovk & Wang, 2021; Ren & Barber, 2024; Li & Zhang, 2025) and applying e-values to ensure offline FDR control (Wang & Ramdas, 2022). In the online testing problem, a related work is Xu & Ramdas (2024), which exploited e-values and proposed the e-LOND algorithm to provide online FDR control under arbitrary, possibly unknown, dependence. However, the e-LOND algorithm does not make full use of the entire error budget and assigns testing levels only by some pre-specified descending sequences related to the number of rejections. Hence, e-LOND yields conservative FDR and sacrifices power, which hampers its practical use. Xu & Ramdas (2024) proposed to improve e-LOND by incorporating independent randomization, though this operation only achieves a smaller improvement in power while introducing additional randomness.

Therefore, a natural question is whether it is possible to construct a GAI-like framework based on e-values to achieve online FDR control under more general dependence, i.e., conditional validity in (2), while efficiently and effectively assigning testing levels to achieve high power.

**Our contributions.** To address this challenge, this paper proposes a novel framework based on e-values, named e-value-based generalized $\alpha$-investing (e-GAI). Our contributions are summarized as follows:

- The e-GAI framework ensures online FDR control based on conditional valid e-values with theoretical guarantees, which is achieved through a new FDP estimator. In contrast to GAI, we propose a novel investing strategy named risk aversion investing (RAI) built on the new FDP estimator, enabling e-GAI to dynamically allocate testing levels based on both prior rejections and costs and assign less $\alpha$-wealth for each rejection to save budget.

- Within the e-GAI framework, we propose two new algorithms called e-LORD and e-SAFFRON. Furthermore, considering the long-term performance, we propose corresponding algorithms, mem-e-LORD and mem-e-SAFFRON, to address the issue of $\alpha$-death, a common phenomenon in the GAI framework.

- Moreover, the e-GAI framework can be generalized to conditionally super-uniform p-values while preserving guaranteed FDR control. Numerical results demonstrate that the algorithms within the e-GAI framework are effective for online FDR control and achieve higher power compared to existing methods.

We compare the e-GAI with several commonly used algorithms and summarize their characteristics in Table 1.

## 2. Preliminaries

### 2.1. p-values & e-values

In this paper, the goal is to make a real-time decision $\delta_t$ while controlling online FDR at a user-specific level $\alpha$. The rejection decision $\delta_t$ with p-values or e-values is defined as, respectively,

$$\delta_t = \begin{cases} \mathbb{1}\left\{p_t \leq \alpha_t\right\}, & \text{if using p-values,} \\ \mathbb{1}\left\{e_t \geq \frac{1}{\alpha_t}\right\}, & \text{if using e-values.} \end{cases}$$

Denote $\mathcal{F}_t = \sigma(\delta_1, \ldots, \delta_t)$ as the sigma-field at time $t$, which is generated by historical decisions in the past. Here the testing level $\alpha_t$ is required to be predictable at time $t$, that is $\alpha_t$ is $\mathcal{F}_{t-1}$-measurable, i.e. $\alpha_t \in \mathcal{F}_{t-1}$.

In the studies of online testing, a valid p-value $p_t$ satisfies the conditionally super-uniform property under the null:

$$\mathbb{P}(p_t \leq u \mid \mathcal{F}_{t-1}) \leq u \text{ for all } u \in [0, 1] \text{ if } \theta_t = 0. \quad (1)$$

Meanwhile, a non-negative variable $e_t$ is a valid e-value if it satisfies the conditional validity:

$$\mathbb{E}[e_t \mid \mathcal{F}_{t-1}] \leq 1 \text{ if } \theta_t = 0. \quad (2)$$

In contrast to the condition on the distribution of a p-value in (1), (2) only requires that the expectation of the e-value

*Table 1.* Online testing algorithms with their properties and performance of FDR control.

| Framework | Algorithm | Statistics | Dependence conditions | $\alpha_t$ relying on prior costs |
|---|---|---|---|---|
| GAI | LORD++ (Ramdas et al., 2017) | p-value | Independence or PRDS | ✔ |
| | SAFFRON (Ramdas et al., 2018) | p-value | Independence or PRDS | ✔ |
| | SupLORD (Xu & Ramdas, 2022) | p-value | (1) | ✔ |
| e-GAI | e-LOND (Xu & Ramdas, 2024) | e-value | Arbitrary dependence | ✗ |
| | **e-LORD** | e-value | (2) | ✔ |
| | **e-SAFFRON** | e-value | (2) | ✔ |

exists and is bounded. This relaxed restriction provides greater flexibility in constructing valid e-values for various practical purposes (Vovk & Wang, 2021; Wang & Ramdas, 2022; Ren & Barber, 2024; Li & Zhang, 2025).

### 2.2. Recap: GAI

The GAI rules are capable of handling an infinite stream of hypotheses and incorporating informative domain knowledge into a dynamic decision-making process. Beginning with a pre-specified $\alpha$-wealth, the key idea in GAI algorithms is that each rejection gains some extra $\alpha$-wealth, which may be subsequently used to make more discoveries at later time points.

Ramdas et al. (2017) provided a statistical perspective on online FDR procedures and proposed to design new algorithms by keeping an estimate of online FDP less than $\alpha$. Specifically, Ramdas et al. (2017) proposed an oracle approximation of online FDP as:

$$\text{FDP}^*(t) = \frac{\sum_{j \in \mathcal{H}_0(t)} \alpha_j}{\left(\sum_{j=1}^t \delta_j\right) \vee 1}. \tag{3}$$

This $\text{FDP}^*(t)$ overestimates the unknown $\text{FDP}(t)$ and provides guidance for online FDR procedures based on independent p-values, including LORD++ (Ramdas et al., 2017), SAFFRON (Ramdas et al., 2018) and ADDIS (Tian & Ramdas, 2019) algorithms.

Specifically, LORD++ (Ramdas et al., 2017) realizes online FDR control by providing a simple upper bound of $\text{FDP}^*(t)$:

$$\widehat{\text{FDP}}^{\text{LORD}}(t) = \frac{\sum_{j=1}^t \alpha_j}{\left(\sum_{j=1}^t \delta_j\right) \vee 1}. \tag{4}$$

If the proportion of alternatives is non-negligible, then LORD++ with $\widehat{\text{FDP}}^{\text{LORD}}(t)$ yields very conservative results due to the overestimation of $\text{FDP}^*(t)$.

Motivated by Storey-BH (Storey, 2002), SAFFRON (Ramdas et al., 2018) was derived from an adaptive upper bound

estimate by approximating the proportions of nulls:

$$\widehat{\text{FDP}}^{\text{SAFFRON}}(t) = \frac{\sum_{j=1}^t \alpha_j \frac{\mathbb{1}\{p_j > \lambda\}}{1-\lambda}}{\left(\sum_{j=1}^t \delta_j\right) \vee 1}, \tag{5}$$

where $\lambda \in (0, 1)$ is a user-chosen parameter.

If the null p-values are independent of each other and of the non-nulls, and $\{\alpha_t\}$ is chosen to be a (coordinate-wise) monotone function of $\boldsymbol{\delta}_{t-1}$, then LORD++ and SAFFRON control the FDR at all times. Fisher (2024) considered the performance of LORD++ and SAFFRON and proved online FDR control when the popular PRDS condition (Benjamini & Yekutieli, 2001) holds. However, the conditions of independence or PRDS are usually violated in practical applications.

## 3. e-GAI: e-value-based GAI

In this section, we first define the oracle estimate of FDP that is tailored for e-values and demonstrate the theoretical results for FDR control (Section 3.1). We then design a new investing strategy based on the new proposed FDP estimator, and propose our testing algorithms, e-LORD (Section 3.2) and e-SAFFRON (Section 3.3), from the risk aversion perspective to optimize the use of a limited budget.

### 3.1. Online FDR Control with e-values

Suppose we observe valid e-values $e_1, \ldots, e_t, \ldots$ and make decision $\delta_t = \mathbb{1}\{e_t \geq 1/\alpha_t\}$ at each time $t$. Inspired by (3), we define an oracle e-value-based estimate of FDP and bound this overestimate to realize FDR control. Denote $R_t = \sum_{j=1}^t \delta_j$ as the rejection size up to $t$.

**Theorem 3.1.** *Suppose the online e-values are valid in* (2). *Let the oracle e-value-based estimate of FDP be given as*

$$\text{FDP}_{\text{e}}^*(t) = \sum_{j \in \mathcal{H}_0(t)} \frac{\alpha_j}{R_{j-1} + 1}. \tag{6}$$

*If* $\mathbb{E}\left[\text{FDP}_{\text{e}}^*(t)\right] \leq \alpha$, *then* $\text{FDR}(t) \leq \alpha$ *for all* $t$.

Theorem 3.1 provides a general theoretical result for FDR control with conditionally valid dependent e-values, guiding

and inspiring the construction of testing levels $\{\alpha_t\}$, which will be detailed in the following subsection. The proof of Theorem 3.1 and any other necessary proofs will be detailed in Appendix A.

Before pursuing further, we discuss the effect of the denominator $R_{j-1}+1$ of $\mathrm{FDP}_e^*(t)$ in (6). To avoid the dependence inflating FDR, the denominator $R_{j-1}+1$ of $\mathrm{FDP}_e^*(t)$ plays an important role in "predicting" the number of possible future rejections at each rejection moment. As the true denominator $R_t \vee 1$ of FDP is unobservable at time $j-1$ and $R_{j-1}+1 \leq (R_t \vee 1)$ holds for each $j \in \{\ell \leq t : \delta_\ell = 1\}$, we use $R_{j-1}+1$ as a substitute that serves as a $(j-1)$-measurable lower bound for $R_t \vee 1$. Since it is placed in the denominator, this results in an overestimation of the oracle FDP, which can subsequently be leveraged to achieve FDR control.

When complex dependence exists, the correlation between the number of false rejections and the total number of rejections cannot be characterized, making it impossible to control their proportion. For instance, consider that the sequential e-values are strongly positively correlated. When a false rejection occurs, it indicates that the next e-value is likely to belong to the null but be falsely rejected as well, leading to an increased FDR. Hence, it implies that one can expect a quantity between $R_{j-1}+1$ and $R_t$ that provides a more efficient approximation for the denominator of $\mathrm{FDP}_e^*(t)$ when knowing some specific dependence structure among e-values, which warrants further study for improving the efficiency of e-GAI. Typically, the denominator of $\mathrm{FDP}_e^*(t)$ in (6) can be directly chosen as $R_t$ for independent e-values; see details in Appendix B.5.

## 3.2. e-LORD

Inspired by the GAI framework, we design testing levels $\alpha_t$ by proposing an upper bound for (6) to realize FDR control according to Theorem 3.1.

One natural overestimate of $\mathrm{FDP}_e^*(t)$ is to define

$$\widehat{\mathrm{FDP}}_e^{\mathrm{LORD}}(t) := \sum_{j=1}^{t} \frac{\alpha_j}{R_{j-1}+1}. \qquad (7)$$

Since $\widehat{\mathrm{FDP}}_e^{\mathrm{LORD}}(t) \geq \mathrm{FDP}_e^*(t)$, any rejection rule algorithm assigning $\alpha_t$ in an online fashion such that $\widehat{\mathrm{FDP}}_e^{\mathrm{LORD}}(t) \leq \alpha$ holds for all $t$ can control online FDR.

**Proposition 3.2.** *Suppose online e-values are valid in* (2). *For* $\alpha_t \in \mathcal{F}_{t-1}$ *satisfying* $\widehat{\mathrm{FDP}}_e^{\mathrm{LORD}}(t) \leq \alpha$, *we have* $\mathrm{FDR}(t) \leq \alpha$ *for all* $t$.

Note that $\widehat{\mathrm{FDP}}_e^{\mathrm{LORD}}(t)$ in (7) is constituted by the summation of terms associated with both $\alpha_t$ and $R_{t-1}$ at each

time point and illustrates that the prior costs associated with each testing (investing) will affect the current test. The target FDR level $\alpha$ is considered as the limited budget ($\alpha$-wealth) of the entire testing procedure, and Proposition 3.2 reveals that it cannot be increased once the testing begins. Unlike GAI's updating strategy, we cannot compensate for $\alpha$-wealth in the subsequent testing process based on the estimate of FDP in e-GAI, as the complex correlations make it difficult to measure the future loss of one false discovery effectively. In contrast, we adopt a *risk aversion* investing (RAI) strategy to update $\alpha_t$ as follows.

Intuitively, one may update testing levels by allocating a prescribed proportion of the remaining budget to satisfy the condition in Proposition 3.2. Therefore, we dynamically allocate testing levels as $\alpha_1 = \alpha\omega_1$ and for $t \geq 2$,

$$\alpha_t = \omega_t \left( \alpha - \sum_{j=1}^{t-1} \frac{\alpha_j}{R_{j-1}+1} \right) (R_{t-1}+1), \qquad (8)$$

where $\omega_1, \ldots, \omega_t \in (0,1)$ control the proportion of the remaining $\alpha$-wealth allocated to the current testing. It's noted that a larger $\omega_t$ indicates that more $\alpha$-wealth is currently invested, which also implies a greater possibility of rejecting the current hypothesis. However, since rejections are not able to gain additional wealth and each test consumes a proportion $\omega_t$ of the remaining wealth, e-GAI views the entire testing process as a risky investment. When failing to reject the current hypothesis, one may consider increasing the investment proportion $\omega_t$ to encourage further testing. Upon hypothesis rejection (deciding to invest), this decision carries both the risk of a false discovery and induces a significant downward bias in the denominator of the FDP estimator at earlier time points. As each rejection introduces new risks akin to an investment, we prioritize updating $\omega_t$ from the RAI perspective as

$$\omega_{t+1} = \omega_t + \omega_1 \varphi^{t-R_t}(1-\delta_t) - \omega_1 \psi^{R_t}\delta_t \qquad (9)$$

$$= \omega_1 + \omega_1 \left( \sum_{j=1}^{t-R_t} \varphi^j - \sum_{j=1}^{R_t} \psi^j \right)$$

with convention $\sum_{j=1}^{0} \varphi^j = \sum_{j=1}^{0} \psi^j = 0$, where $\omega_1 \in (0,1)$ is a user-defined initial allocation coefficient, and $\varphi > 0$, $\psi > 0$ are user-defined parameters that characterize the investment stimulation intensity post-acceptance and the risk regulation level post-rejection, respectively.

*Remark* 3.3. To ensure that $\omega_t \in (0,1)$ for each time $t$, we can select any $\omega_1 \in (0, 0.5)$, $\varphi \in [0, 0.5]$, and $\psi \in [0, 0.5]$. In fact, the conditions can be relaxed to ensure that $\omega_t$ is $\mathcal{F}_{t-1}$-measurable and $\omega_t \in (0,1)$. We suggest choosing $\omega_1 = O(1/T)$ with the total number of hypotheses $T$ to avoid spending too much wealth in the early stages while retaining sufficiently effective wealth for testing at each

---

**Algorithm 1** e-LORD

---

1: **Input:** target FDR level $\alpha$, initial allocation coefficient $\omega_1 \in (0, 1)$, parameters $\varphi$ and $\psi \in (0, 1)$, sequence of e-values $e_1, e_2, \ldots$.

2: Calculate $\alpha_1 = \alpha\omega_1$ and decide $\delta_1 = \mathbb{1}\left\{e_1 \geq \frac{1}{\alpha_1}\right\}$;

3: Update $R_1 = \delta_1$ and $\omega_2$ by (9);

4: **for** $t = 2, 3, \ldots$ **do**

5:     Update testing level $\alpha_t$ by (8);

6:     Make decision $\delta_t = \mathbb{1}\left\{e_t \geq \frac{1}{\alpha_t}\right\}$;

7:     Update $R_t = R_{t-1} + \delta_t$ and $\omega_{t+1}$ by (9);

8: **end for**

9: **Output:** decision set $\{\delta_1, \delta_2, \ldots\}$.

---

time point; more detailed discussions are provided in Appendix B.2. Note that the choice of $\omega_t$ satisfying the above conditions does not affect the guarantee of the FDR control. This observation opens up greater flexibility, enhances the applicability of our algorithms, and enables users to leverage domain knowledge for dynamically adjusting the allocation.

The whole algorithm is referred to as e-LORD and summarized in Algorithm 1. In e-LORD, the testing levels $\{\alpha_t\}$ are updated not only by relying on both the number of previous rejections and the prior costs, but also by assigning less $\alpha$-wealth for each rejection using the RAI strategy.

We find that the e-LOND algorithm proposed by Xu & Ramdas (2024) can be converted into the e-LORD algorithm. Let $\alpha_t^{\text{e-LOND}}$ and $R_t^{\text{e-LOND}}$ denote the testing level and the number of rejections at time $t$ in the e-LOND algorithm, respectively. Xu & Ramdas (2024) assigned $\alpha_t^{\text{e-LOND}} = \alpha\gamma_t \left(R_{t-1}^{\text{e-LOND}} + 1\right)$, where $\{\gamma_t\}$ is pre-specified non-negative sequence summing to one. It can be verified that $\widehat{\text{FDP}}_{\text{e}}^{\text{LORD}}(t)$ in (7) satisfies

$$\widehat{\text{FDP}}_{\text{e}}^{\text{LORD}}(t) = \sum_{j=1}^{t} \frac{\alpha_j^{\text{e-LOND}}}{R_{j-1}^{\text{e-LOND}} + 1} = \alpha \sum_{j=1}^{t} \gamma_t = \alpha.$$

Furthermore, if choosing $\gamma_t = \omega_t \prod_{j=1}^{t-1}(1 - \omega_j)$ in e-LOND with $\{\omega_t\}$ in e-LORD, then we have $\alpha_t^{\text{e-LOND}}$ equals $\alpha_t$ in (8) for any $t$; refer to (18) in Appendix B.6 for more details. In this case, at each time $t$, the rejection set of e-LOND will be identical to the result of Algorithm 1. Thus, we can consider e-LOND as operating on a special type of e-LORD by designing $\omega_t$ from a given sequence $\{\gamma_t\}$.

### 3.3. e-SAFFRON

To approximate $\text{FDP}_{\text{e}}^*(t)$, the estimate $\widehat{\text{FDP}}_{\text{e}}^{\text{LORD}}(t)$ is calculated by summing all non-negative terms over time. Thus, it serves as a crude and conservative overestimate if the proportion of alternatives is non-negligible. Inspired by Storey-BH (Storey, 2002) and SAFFRON (Ramdas et al.,

2018), we further propose an adaptive estimate defined as

$$\widehat{\text{FDP}}_{\text{e}}^{\text{SAFFRON}}(t) := \sum_{j=1}^{t} \frac{\alpha_j}{R_{j-1} + 1} \frac{\mathbb{1}\left\{e_j < \frac{1}{\lambda_j}\right\}}{1 - \lambda_j},$$

where $\{\lambda_t\}_{t=1}^{\infty}$ is a predictable sequence of user-chosen parameters in the interval $(0, 1)$. Here the term *adaptive* means that it is based on an estimate of the proportion of true nulls as in Storey (2002); Ramdas et al. (2018).

In contrast to $\widehat{\text{FDP}}_{\text{e}}^{\text{LORD}}(t)$, the summation in $\widehat{\text{FDP}}_{\text{e}}^{\text{SAFFRON}}(t)$ includes only those test levels associated with relatively small e-values. Although $\widehat{\text{FDP}}_{\text{e}}^{\text{SAFFRON}}(t)$ is not necessarily always larger than $\text{FDP}_{\text{e}}^*(t)$, we can verify that $\mathbb{E}\left[\widehat{\text{FDP}}_{\text{e}}^{\text{SAFFRON}}(t)\right] \geq \mathbb{E}\left[\text{FDP}_{\text{e}}^*(t)\right]$, which is sufficient for FDR control according to Theorem 3.1. The properties of the adaptive estimate are formalized below.

**Proposition 3.4.** *Given a predictable sequence $\{\lambda_t\}_{t=1}^{\infty}$, if online e-values are valid in* (2)*, then for $\alpha_t \in \mathcal{F}_{t-1}$ satisfying $\widehat{\text{FDP}}_{\text{e}}^{\text{SAFFRON}}(t) \leq \alpha$, we have:*

*(a)* $\mathbb{E}\left[\widehat{\text{FDP}}_{\text{e}}^{\text{SAFFRON}}(t)\right] \geq \mathbb{E}\left[\text{FDP}_{\text{e}}^*(t)\right]$ *and*

*(b)* $\text{FDR}(t) \leq \alpha$ *for all $t$.*

In the following, we consider $\lambda_t \equiv \lambda \in (0, 1)$ for simplicity. Embracing the RAI principle, we propose an adaptive algorithm, called e-SAFFRON. The e-SAFFRON allocates testing levels as $\alpha_1 = \alpha(1 - \lambda)\omega_1$ and for $t \geq 2$,

$$\alpha_t = \omega_t \left(\alpha(1 - \lambda) - \sum_{j=1}^{t-1} \frac{\alpha_j \mathbb{1}\left\{e_j < \frac{1}{\lambda}\right\}}{R_{j-1} + 1}\right)(R_{t-1} + 1),$$
(10)

where $\omega_t \in (0, 1)$ is updated by (9).

We summarize e-SAFFRON in Algorithm 2. In particular, setting $\lambda = 0$ in Algorithm 2 simplifies it to Algorithm 1, demonstrating that e-SAFFRON serves as the adaptive counterpart to e-LORD, similar to the relationship between SAFFRON and LORD++ within the GAI framework.

*Remark* 3.5. Note that the choice of $\lambda$ will affect the total "wealth" $\alpha(1 - \lambda)$, which will be no further increased in the subsequent period of e-SAFFRON. We prefer a relatively small value $\lambda$ to preserve wealth, with $\lambda = 0.1$ as the default choice in our numerical experiments, which differs from the value recommended in the SAFFRON procedure (Ramdas et al., 2018). The latter, SAFFRON with independent p-values, allows for additional rewards when a hypothesis is rejected and suggests $\lambda = 0.5$.

---

**Algorithm 2** e-SAFFRON
1: **Input:** target FDR level $\alpha$, initial allocation coefficient $\omega_1 \in (0,1)$, parameters $\lambda, \varphi$ and $\psi \in (0,1)$, sequence of e-values $e_1, e_2, \ldots$.
2: Calculate $\alpha_1 = \alpha(1-\lambda)\omega_1$ and decide $\delta_1 = \mathbb{1}\left\{e_1 \geq \frac{1}{\alpha_1}\right\}$;
3: Update $R_1 = \delta_1$ and $\omega_2$ by (9);
4: **for** $t = 2, 3, \ldots$ **do**
5:     Update testing level $\alpha_t$ by (10);
6:     Make decision $\delta_t = \mathbb{1}\left\{e_t \geq \frac{1}{\alpha_t}\right\}$;
7:     Update $R_t = R_{t-1} + \delta_t$ and $\omega_{t+1}$ by (9);
8: **end for**
9: **Output:** decision set $\{\delta_1, \delta_2, \ldots\}$.

---

# 4. Further Discussions on e-GAI

In this section, we further investigate the properties of the e-LORD and e-SAFFRON algorithms within the e-GAI framework.

## 4.1. Long-Term Performance

We provide strategies for the long-term performance of our methods to address the issue of $\alpha$-death, halting rejections once $\alpha$-wealth tends to zero, a common phenomenon within the GAI framework (Ramdas et al., 2017).

$\alpha$**-death.**   In a long-term testing process, there may be extended periods during which no hypotheses are rejected, particularly when the true alternatives are rare, leading to a continuous accumulation of the allocation proportion $\omega_t$. As a result, testing levels may become severely diminished in the later stages, making it difficult to achieve any further rejections. This phenomenon is referred to as $\alpha$-death, which induces a loss of power and ultimately compromises the long-term efficacy of our online testing algorithm.

**mem-FDR control.**   To alleviate $\alpha$-death over a long period (potentially infinite), Ramdas et al. (2017) defined decaying memory FDR (mem-FDR) to allow more attention to recent rejections by introducing a user-defined decay parameter $d \in (0,1]$ and proposed mem-LORD++ that controls mem-FDR under independence. Specifically, mem-FDR is defined as

$$\text{mem-FDR}(t) := \mathbb{E}\left[\frac{\sum_{j \in \mathcal{H}_0(t)} d^{t-j}\delta_j}{\sum_{j=1}^{t} d^{t-j}\delta_j}\right].$$

To address the issue, we adapt e-GAI and design mem-e-GAI to control mem-FDR in our setting. The technique used here is similar to the e-GAI framework to control FDR in Section 3. Denote $R_t^{\text{d}} = \sum_{j=1}^{t} d^{t-j}\delta_j$ for simplicity.

**Theorem 4.1.** *Suppose the online e-values are valid in* (2). *Let the oracle e-value-based estimate of mem-FDP be*

$$\text{mem-FDP}^*(t) := \sum_{j \in \mathcal{H}_0(t)} \frac{\alpha_j}{dR_{j-1}^{\text{d}} + 1}. \qquad (11)$$

*If* $\mathbb{E}\left[\text{mem-FDP}^*(t)\right] \leq \alpha$, *then* $\text{mem-FDR}(t) \leq \alpha$ *for all* $t$.

Theorem 4.1 provides an oracle estimate of mem-FDP, and offers insights and guidance for designing algorithms that control mem-FDR. Note that (11) is facilitated by an understanding of the unknown denominator $R_t^{\text{d}}$ of the true mem-FDP. A natural choice that can serve as a $(j-1)$-measurable lower bound for *predicting* $R_t^{\text{d}}$ is $d^{t-j}\left(dR_{j-1}^{\text{d}} + 1\right)$ since this predicted value $d^{t-j}\left(dR_{j-1}^{\text{d}} + 1\right) \leq \left(R_t^{\text{d}} \vee 1\right)$ holds for each $j$ with $\delta_j = 1$.

**mem-e-LORD & mem-e-SAFFRON.**   Adopting the core idea of the e-GAI framework, we can construct upper bounds for $\text{mem-FDP}^*(t)$ in (11) and design the testing levels accordingly to achieve mem-FDR control.

One natural overestimate of $\text{mem-FDP}^*(t)$ is

$$\text{mem-}\widehat{\text{FDP}}^{\text{LORD}}(t) := \sum_{j=1}^{t} \frac{\alpha_j}{dR_{j-1}^{\text{d}} + 1} \qquad (12)$$

and any algorithm is referred to as mem-e-LORD that allocates testing levels $\{\alpha_t\}$ satisfying $\text{mem-}\widehat{\text{FDP}}^{\text{LORD}}(t) \leq \alpha$. As an example, adopting the RAI strategy, mem-e-LORD allocates testing levels as $\alpha_1 = \alpha\omega_1$ and for $t \geq 2$,

$$\alpha_t = \omega_t\left(\alpha - \sum_{j=1}^{t-1} \frac{\alpha_j}{dR_{j-1}^{\text{d}} + 1}\right)\left(dR_{t-1}^{\text{d}} + 1\right), \quad (13)$$

where $\omega_t \in (0,1)$ is updated by (9).

When the proportion of alternatives is non-negligible, it is preferable to employ an adaptive overestimate defined as

$$\text{mem-}\widehat{\text{FDP}}^{\text{SAFFRON}}(t) := \sum_{j=1}^{t} \frac{\alpha_j}{dR_{j-1}^{\text{d}} + 1} \frac{\mathbb{1}\left\{e_j < \frac{1}{\lambda_j}\right\}}{1 - \lambda_j},$$

where $\{\lambda_t\}_{t=1}^{\infty}$ satisfying $\lambda_t \in (0,1)$ is a predictable sequence of user-chosen. We refer to an algorithm as mem-e-SAFFRON that allocates testing levels $\{\alpha_t\}$ satisfying $\text{mem-}\widehat{\text{FDP}}^{\text{SAFFRON}}(t) \leq \alpha$. For simplicity, we consider $\lambda_t \equiv \lambda \in (0,1)$ and employ mem-e-SAFFRON from RAI perspective, allocating testing levels as $\alpha_1 = \alpha(1-\lambda)\omega_1$ and for $t \geq 2$,

$$\alpha_t = \omega_t\left(\alpha(1-\lambda) - \sum_{j=1}^{t-1} \frac{\alpha_j \mathbb{1}\left\{e_j < \frac{1}{\lambda}\right\}}{dR_{j-1}^{\text{d}} + 1}\right)\left(dR_{t-1}^{\text{d}} + 1\right),$$

and updating $\omega_t \in (0, 1)$ as in (9).

According to Theorem 4.1, both mem-e-LORD and mem-e-SAFFRON achieve mem-FDR control.

**Proposition 4.2.** *Suppose online e-values are valid in* (2).

(a) *For* $\alpha_t \in \mathcal{F}_{t-1}$ *satisfying* mem-$\widehat{\text{FDP}}^{\text{LORD}}(t) \leq \alpha$, *we have* mem-FDR$(t) \leq \alpha$ *for all* $t$.

(b) *Given a predictable sequence* $\{\lambda_t\}_{t=1}^{\infty}$, *for* $\alpha_t \in \mathcal{F}_{t-1}$ *satisfying* mem-$\widehat{\text{FDP}}^{\text{SAFFRON}}(t) \leq \alpha$, *we have* $\mathbb{E}\left[\text{mem-}\widehat{\text{FDP}}^{\text{SAFFRON}}(t)\right] \geq \mathbb{E}\left[\text{mem-FDP}^*(t)\right]$ *and* mem-FDR$(t) \leq \alpha$ *for all* $t$.

### 4.2. Extension to p-values

While the e-GAI framework is initially developed based on the study of e-values, we demonstrate that it can also be generalized to p-values satisfying conditionally super-uniformity in (1), enriching the proposed framework and making the theory more comprehensive and complete.

Suppose we observe valid p-values $p_1, \ldots, p_t, \ldots$ and make decision $\delta_t = \mathbb{1}\{p_t \leq \alpha_t\}$ at each time $t$. When using p-values for online testing, we demonstrate that $\text{FDP}_{\text{e}}^*(t)$ in (6) can still serve as an oracle estimate of FDP, in which the denominator involves the number of rejections $R_t$ relevant to p-values. By controlling this estimator to be bounded, we can achieve FDR control for p-values satisfying the conditionally super-uniform property in (1).

**Theorem 4.3.** *Suppose the online p-values are conditionally super-uniform in* (1). *If* $\text{FDP}_{\text{e}}^*(t)$ *in* (6) *satisfies* $\mathbb{E}\left[\text{FDP}_{\text{e}}^*(t)\right] \leq \alpha$, *then* FDR$(t) \leq \alpha$ *for all* $t$.

Theorem 4.3 provides a general theoretical result for FDR control with conditionally super-uniform p-values. Building upon the analogous strategy outlined in Section 3 and Section 4.1, the e-GAI framework can be naturally extended to p-value-compatible algorithms and corresponding versions for the long-term performance. For precise algorithmic differentiation, we denote the p-value-adapted variants of e-LORD and e-SAFFRON as pL-RAI and pS-RAI, respectively, emphasizing the adoption of p-values as the test statistics and the dynamic updating mechanism of testing levels $\alpha_t$ through the RAI strategy. We provide more detailed discussions and technical explanations in Appendix B.4.

## 5. Numerical Experiments

In this section, we evaluate the performance of our online testing framework on both synthetic and real data. We compare e-LORD, e-SAFFRON, pL-RAI, and pS-RAI with e-LOND, LORD++, SAFFRON, and SupLORD in terms of FDR and power. We validate the performance

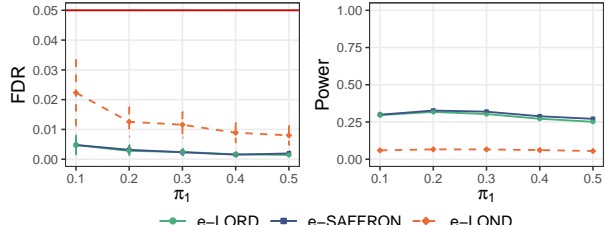

(a) e-value-based methods

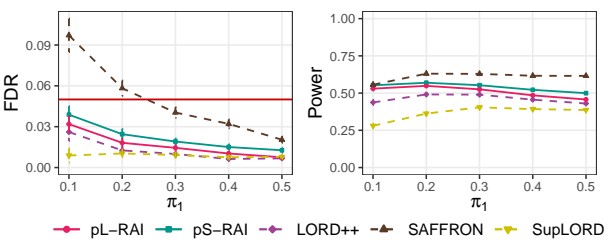

(b) p-value-based methods

*Figure 1.* Empirical FDR and power with standard error versus proportion of alternative hypotheses $\pi_1$ for various online methods, with $\rho = 0.5$, $L = 30$ and $\mu_c = 3$.

of mem-e-LORD and mem-e-SAFFRON through simulated numerical experiments in Appendix C.2. The code for all numerical experiments in this paper is available at https://github.com/zijianwei01/e-GAI.

### 5.1. Simulation: Testing with Gaussian Observations

We use an experimental setup that tests the mean of a Gaussian distribution with the total number of data $T = 500$. The null hypothesis takes $\mathbb{H}_t : \mu_t = 0$ for each time $t \in [T]$. The true labels $\theta_t$ are generated from Bernoulli($\pi_1$). The Gaussian variates $(X_1, \ldots, X_T)^\top$ is from $\mathcal{N}(\boldsymbol{\mu}, \boldsymbol{\Sigma})$ with mean vector $\boldsymbol{\mu} \in \mathbb{R}^T$ and covariance matrix $\boldsymbol{\Sigma} \in \mathbb{R}^{T \times T}$. The elements in $\boldsymbol{\mu} = (\mu_1, \ldots, \mu_T)^\top$ satisfy $\mu_t = 0$ if $\theta_t = 0$ and $\mu_t = \mu_c > 0$ if $\theta_t = 1$, where $\mu_c$ is the signal parameter. Additionally, the signals of the true alternatives are correlated with the correlation coefficient $\rho$. The covariance matrix satisfies $\Sigma \succ 0$, and $\Sigma_{ij} = \rho^{|i-j|} \cdot \mathbb{1}\{|i-j| \leq L\}$. Note that the data at different time points will influence each other, which is in line with real-life online scenarios.

Under the normality assumption at each time point $t$, we compute the e-value as the corresponding likelihood-ratio statistic and the p-value by evaluating the conditional distribution. We take $\omega_1 = 0.005, \varphi = \psi = 0.5$ in e-LORD and pL-RAI, and additionally $\lambda = 0.1$ in e-SAFFRON and pS-RAI, while we use default parameters from the R package onlineFDR (Robertson et al., 2022) for other benchmarks. The target FDR level is set as $\alpha = 0.05$.

*Table 2.* Proportion of points rejected out of anomalous regions. e-LORD, e-SAFFRON and e-LOND control the estimated FDP under $\alpha = 0.1$. e-SAFFRON loses some power due to a tiny proportion of true alternatives.

| Method | e-LORD | e-SAFFRON | e-LOND |
|---|---|---|---|
| $\widehat{\text{FDP}}$ | 0.085 | 0.087 | 0.061 |
| Num Discovery | 47 | 46 | 33 |

Figure 1 show the results of various methods with $\rho = 0.5$ and $\mu_c = 3$. The empirical results show that SAFFRON will inflate the FDR heavily, while others realize the FDR control. However, there is no theoretical guarantee for controlling the FDR with complex dependent data in LORD++, which makes the rejection decisions not as safe as they seem. Owing to dynamically updating the testing levels, both e-LORD and e-SAFFRON lead to much higher power than e-LOND, and pL-RAI and pS-RAI lead to higher power than LORD++ and SupLORD. Similar performance can be found for other settings, as presented in Appendix C.1.

### 5.2. Real Data: NYC Taxi Anomaly Detection

We analyze the NYC taxi dataset from the Numenta Anomaly Benchmark (NAB) repository (Lavin & Ahmad, 2015). The dataset captures the number of NYC taxi passengers every 30 minutes from July 1, 2014, to January 31, 2015. Five known anomalous intervals correspond to notable events such as the NYC marathon, Thanksgiving, Christmas, New Year's Day, and a snowstorm. We visualize the data with the known anomalous intervals highlighted using red rectangles in Figure 2. The anomaly detection problem is formulated as an online multiple testing problem.

We employ the R package `stlplus` to perform STL decomposition (Cleveland et al., 1990) to remove the seasonal and trend components. We derive tests on the residuals, which are assumed to form an independent sequence. The first 2000 time points are taken as the initial sequence for model calibration. We focus on the comparisons among e-value-based methods and apply e-LORD, e-SAFFRON and e-LOND to analyze this dataset. We use the estimated likelihood ratio as e-values (shown in (21)). We choose $\omega_1 = 0.0001$ and $\lambda = 0.1$ and set both $\psi$ and $\varphi$ as 0.5.

We compare their performance in terms of the proportion of discoveries out of marked anomalous regions, denoted here as $\widehat{\text{FDP}}$, and the number of discovered anomalous regions in Table 2. Our e-GAI procedures effectively maintain $\widehat{\text{FDP}}$ below the target level. As illustrated in Figure 2, e-LORD and e-SAFFRON demonstrate higher power than e-LOND, as both identify more points within the anomalous regions, shown in red squares. More comparisons for p-value-based methods are shown in Appendix C.3.

### 5.3. Real Data: Dating Financial Bubbles

We follow Genoni et al. (2023), building a sequential test on stock market prices for financial bubbles. Online testing procedures enable decision-making regarding bubble occurrence based on current observation before the subsequent one is observed. Controlling FDR is a proper guarantee to make decisions with a controllable proportion of mistakes.

The analysis is performed on the monthly stock price of the Nasdaq series. The calibration is implemented by using the first $1/3$ observations, assuming the related period to be free of bubbles. Genoni et al. (2023) uses the standard p-values of the ADF test (R package `urca`), which are calculated point-wise and thus do not satisfy the conditionally super-uniform property. Moreover, such a valid p-value is difficult to construct for time series. Building on Dickey & Fuller (1981), we reformulate the likelihood ratio for the unit root test as sequential e-values and further normalize them by the estimated conditional expectation under the null to guarantee conditional validity. The task is to identify the bubble beginning date (BBD) and bubble ending date (BED).

As shown in Figure 3, with a significance level of $\alpha = 0.005$, e-LOND only detects the initial potential change associated with the gradual emergence of technology companies in the marketplace. The e-LORD and e-SAFFRON exhibit similar behavior, providing a comprehensive characterization of the entire potential bubble influence region by marking a dense rejection region. The date 1990-01-20, corresponding to the onset of the bubble, has been identified, aligning with the classical view. The BED in Figure 3 is determined based on empirical experience due to the long-term effects of the dot-com bubble burst. Specifically, while the burst of the dot-com bubble led to a reset in valuations, NASDAQ's volatility remained elevated, exceeding pre-bubble levels and establishing the groundwork for subsequent financial cycles. As a result, this sustained volatility led to persistent rejections in online procedures even after BBD. The region detected by e-LORD also includes other historical events after BED.

### 6. Summary

In this paper, we propose a novel framework named e-GAI that introduces an oracle estimate of online FDP and ensures online FDR control under conditional validity with theoretical guarantees. The e-GAI dynamically allocates testing levels from the RAI perspective, which relies on both the number of previous rejections and prior costs and assigns less $\alpha$-wealth for each rejection. Within the e-GAI framework, we propose two new algorithms, e-LORD and e-SAFFRON, respectively. Both e-LORD and e-SAFFRON are more powerful than e-LOND under complicated dependence. We also

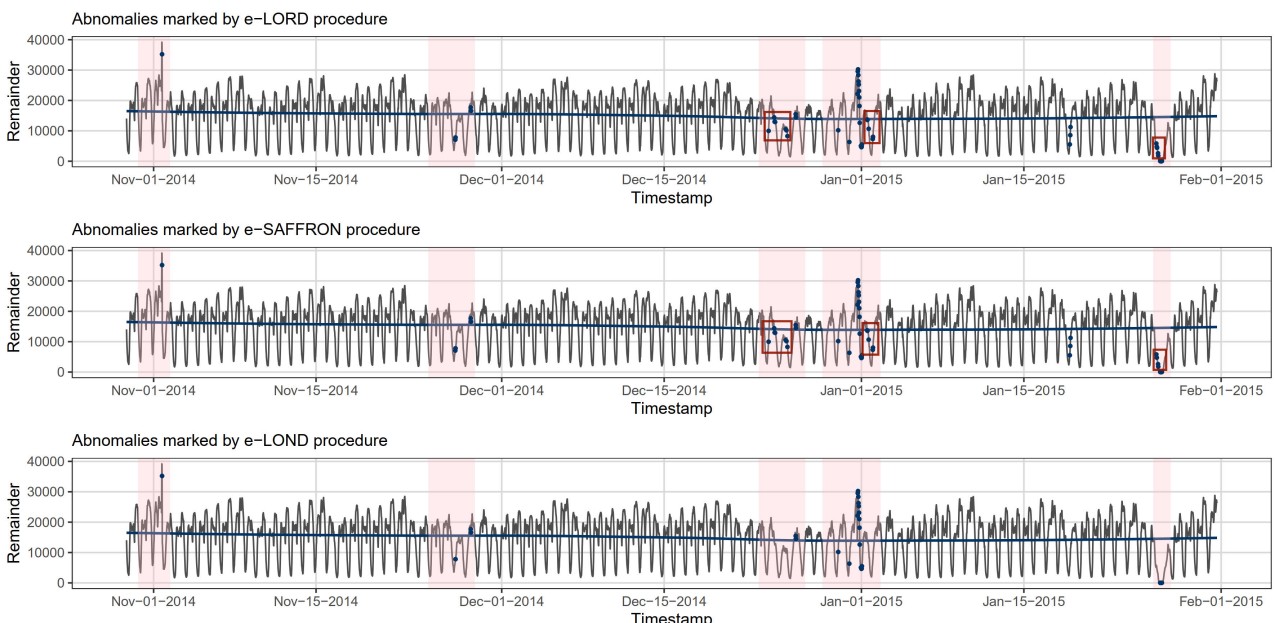

Figure 2. Anomaly points detected by e-LORD (above), e-SAFFRON (middle) and e-LOND (below). Rejection points of all procedures are marked by dark blue points. Red regions refer to known anomalies. The testing level is chosen as 0.1. Red squares indicate additional discovery of e-LORD and e-SAFFRON compared to e-LOND.

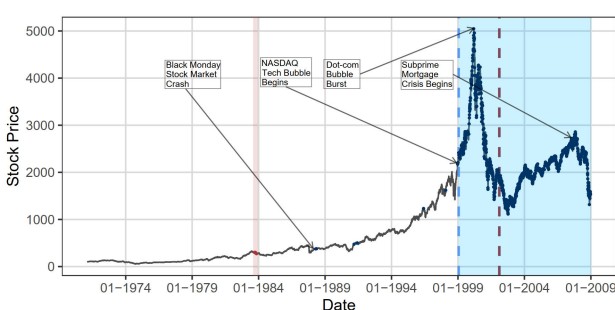

Figure 3. Dating of BBD and BED in the Nasdaq series. Blue and red points refer to e-LORD and e-LOND rejection points under $\alpha = 0.005$, respectively. Blue and red regions refer to bubble influence regions detected by e-LORD and e-LOND. Blue and red dashed lines refer to BBD and BED marked by the Nasdaq series. Significant historical events are marked.

propose mem-e-LORD and mem-e-SAFFRON correspondingly for the long-term performance to alleviate $\alpha$-death. Moreover, we demonstrate that e-GAI can be generalized to conditionally super-uniform p-values, making e-GAI a more versatile tool with reliable theoretical guarantees and increased practical value.

We conclude this work with two remarks. Firstly, although

e-SAFFRON provides an adaptive online FDR procedure, how to more accurately approximate the proportions of nulls like ADDIS (Tian & Ramdas, 2019) deserves further research. Secondly, as discussed in Section 3.1, a more accurate approximation of $\mathrm{FDP}_e^*(t)$ can be expected when knowing a more specific dependence structure. It warrants future study on the relationship between the denominator of $\mathrm{FDP}_e^*(t)$ and the dependence structure, and how to design a more efficient strategy to assign testing levels under such cases.

## Acknowledgements

We sincerely thank the anonymous reviewers for their insightful comments and constructive suggestions, which have greatly improved the quality of this manuscript. Haojie Ren was supported by the National Key R&D Program of China (Grant No. 2024YFA1012200), the National Natural Science Foundation of China (Grant No. 12471262), the Young Elite Scientists Sponsorship Program by CAST and Shanghai Jiao Tong University 2030 Initiative. Changliang Zou was supported by the National Key R&D Program of China (Grant Nos. 2022YFA1003703, 2022YFA1003800) and the National Natural Science Foundation of China (Grant No. 12231011).

## Impact Statement

This paper presents work whose goal is to advance the field of Machine Learning. There are many potential societal consequences of our work, none which we feel must be specifically highlighted here.

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

## A. Proofs

Here, we include all the proofs of the results throughout the paper.

### A.1. Proof of Theorem 3.1

*Proof.* Suppose a desired level $\alpha$ is given. For all time $t$, we have

$$
\begin{aligned}
\mathrm{FDR}(t) =& \mathbb{E}\left[\frac{\sum_{j\in\mathcal{H}_0(t)}\delta_j}{R_t\vee 1}\right] \overset{(i)}{\leq} \mathbb{E}\left[\sum_{j\in\mathcal{H}_0(t)}\frac{\delta_j}{R_{j-1}+1}\right] = \mathbb{E}\left[\sum_{j\in\mathcal{H}_0(t)}\frac{\mathbb{1}\{e_j\geq\frac{1}{\alpha_j}\}}{R_{j-1}+1}\right] \overset{(ii)}{\leq} \mathbb{E}\left[\sum_{j\in\mathcal{H}_0(t)}\frac{e_j\alpha_j}{R_{j-1}+1}\right] \\
=& \mathbb{E}\left[\sum_{j\in\mathcal{H}_0(t)}\frac{\mathbb{E}\left[e_j\mid\mathcal{F}_{j-1}\right]\alpha_j}{R_{j-1}+1}\right] \overset{(iii)}{\leq} \mathbb{E}\left[\sum_{j\in\mathcal{H}_0(t)}\frac{\alpha_j}{R_{j-1}+1}\right] = \mathbb{E}\left[\mathrm{FDP}_{\mathrm{e}}^*(t)\right] \overset{(iv)}{\leq} \alpha,
\end{aligned}
$$

where the inequality (i) holds since $R_{j-1}+1 \leq (R_t\vee 1)$ for every $j\in\{j\leq t:\delta_j=1\}$ by definition, the inequality (ii) holds since $\mathbb{1}\{y\geq 1\}\leq y$ for any $y>0$, the inequality (iii) follows after taking iterated expectations by conditioning on $\mathcal{F}_{j-1}$ and then applying the property of e-values, and inequality (iv) holds by condition. Thus, we finish the whole proof. $\qquad\square$

### A.2. Proof of Proposition 3.2

*Proof.* To prove the property, we only need to verify that $\mathbb{E}\left[\widehat{\mathrm{FDP}}_{\mathrm{e}}^{\mathrm{LORD}}(t)\right] \geq \mathbb{E}\left[\mathrm{FDP}_{\mathrm{e}}^*(t)\right]$ for all $t$, then we can obtain the desired result by Theorem 3.1. This holds trivially since $\widehat{\mathrm{FDP}}_{\mathrm{e}}^{\mathrm{LORD}}(t)\geq\mathrm{FDP}_{\mathrm{e}}^*(t)$ by construction. $\qquad\square$

### A.3. Proof of Proposition 3.4

*Proof.* Given a desired level $\alpha$ and predictable sequence $\{\lambda_t\}_{t=1}^\infty$, for all $t$,

$$
\begin{aligned}
\mathbb{E}\left[\widehat{\mathrm{FDP}}_{\mathrm{e}}^{\mathrm{SAFFRON}}(t)\right] =& \sum_{j=1}^t \mathbb{E}\left[\frac{\alpha_j}{R_{j-1}+1}\frac{\mathbb{1}\left\{e_j<\frac{1}{\lambda_j}\right\}}{1-\lambda_j}\right] \\
\geq& \sum_{j\in\mathcal{H}_0(t)}\mathbb{E}\left[\frac{\alpha_j}{R_{j-1}+1}\frac{\mathbb{1}\left\{e_j<\frac{1}{\lambda_j}\right\}}{1-\lambda_j}\right] \\
=& \sum_{j\in\mathcal{H}_0(t)}\mathbb{E}\left[\frac{\alpha_j}{R_{j-1}+1}\frac{\mathbb{E}\left[\mathbb{1}\left\{e_j<\frac{1}{\lambda_j}\right\}\mid\mathcal{F}_{j-1}\right]}{1-\lambda_j}\right] \\
\overset{(i)}{\geq}& \sum_{j\in\mathcal{H}_0(t)}\mathbb{E}\left[\frac{\alpha_j}{R_{j-1}+1}\right] = \mathbb{E}\left[\mathrm{FDP}_{\mathrm{e}}^*(t)\right],
\end{aligned}
$$

where inequality (i) holds because $\mathbb{E}\left[\mathbb{1}\left\{e_j<\frac{1}{\lambda_j}\right\}\mid\mathcal{F}_{j-1}\right]\geq 1-\lambda_j$ by the property of e-values. This concludes the proof of part (a), and then (b) can be derived by Theorem 3.1, which completes the proof. $\qquad\square$

## A.4. Proof of Theorem 4.1

*Proof.* Suppose a desired level $\alpha$ is given. For all time $t$, we have

$$
\begin{aligned}
\text{mem-FDR}(t) =& \mathbb{E}\left[\frac{\sum_{j \in \mathcal{H}_0(t)} d^{t-j} \delta_j}{R_t^{\mathrm{d}} \vee 1}\right] \overset{(i)}{\leq} \mathbb{E}\left[\sum_{j \in \mathcal{H}_0(t)} \frac{d^{t-j} \delta_j}{d^{t-j}(dR_{j-1}^{\mathrm{d}}+1)}\right] = \mathbb{E}\left[\sum_{j \in \mathcal{H}_0(t)} \frac{\mathbb{1}\{e_j \geq \frac{1}{\alpha_j}\}}{dR_{j-1}^{\mathrm{d}}+1}\right] \\
\overset{(ii)}{\leq}& \mathbb{E}\left[\sum_{j \in \mathcal{H}_0(t)} \frac{e_j \alpha_j}{dR_{j-1}^{\mathrm{d}}+1}\right] = \mathbb{E}\left[\sum_{j \in \mathcal{H}_0(t)} \frac{\mathbb{E}\left[e_j \mid \mathcal{F}_{j-1}\right] \alpha_j}{dR_{j-1}^{\mathrm{d}}+1}\right] \overset{(iii)}{\leq} \mathbb{E}\left[\sum_{j \in \mathcal{H}_0(t)} \frac{\alpha_j}{dR_{j-1}^{\mathrm{d}}+1}\right] \\
=& \mathbb{E}\left[\text{mem-FDP}^*(t)\right] \overset{(iv)}{\leq} \alpha,
\end{aligned}
$$

where the inequality (i) holds since $R_t^{\mathrm{d}} = \sum_{k=1}^{j-1} d^{t-k} \delta_k + d^{t-j} \delta_j + \sum_{k=j+1}^{t} d^{t-k} \delta_k \geq d^{t-j+1} R_{j-1}^{\mathrm{d}} + d^{t-j} \delta_j$ and hence $d^{t-j}(dR_{j-1}+1) \leq (R_t^{\mathrm{d}} \vee 1)$ for every $j \in \{j \leq t : \delta_j = 1\}$ by definition, the inequality (ii) holds since $\mathbb{1}\{y \geq 1\} \leq y$ for any $y > 0$, the inequality (iii) follows after taking iterated expectations by conditioning on $\mathcal{F}_{j-1}$ and then applying the property of e-values, and inequality (iv) holds by condition. Thus, we finish the whole proof. $\square$

## A.5. Proof of Proposition 4.2

*Proof.* (a) To prove the property, we only need to verify that $\mathbb{E}\left[\text{mem-}\widehat{\text{FDP}}^{\text{LORD}}(t)\right] \geq \mathbb{E}\left[\text{mem-FDP}^*(t)\right]$ for all $t$, then

we can obtain the desired result by Theorem 4.1. This holds trivially since $\text{mem-}\widehat{\text{FDP}}^{\text{LORD}}(t) \geq \text{mem-FDP}^*(t)$ by construction.

(b) Given a desired level $\alpha$ and predictable sequence $\{\lambda_t\}_{t=1}^{\infty}$, for all $t$,

$$
\begin{aligned}
\mathbb{E}\left[\text{mem-}\widehat{\text{FDP}}^{\text{SAFFRON}}(t)\right] =& \sum_{j=1}^{t} \mathbb{E}\left[\frac{\alpha_j}{dR_{j-1}^{\mathrm{d}}+1} \frac{\mathbb{1}\left\{e_j < \frac{1}{\lambda_j}\right\}}{1-\lambda_j}\right] \\
\geq& \sum_{j \in \mathcal{H}_0(t)} \mathbb{E}\left[\frac{\alpha_j}{dR_{j-1}^{\mathrm{d}}+1} \frac{\mathbb{1}\left\{e_j < \frac{1}{\lambda_j}\right\}}{1-\lambda_j}\right] \\
=& \sum_{j \in \mathcal{H}_0(t)} \mathbb{E}\left[\frac{\alpha_j}{dR_{j-1}^{\mathrm{d}}+1} \frac{\mathbb{E}\left[\mathbb{1}\left\{e_j < \frac{1}{\lambda_j}\right\} \mid \mathcal{F}_{j-1}\right]}{1-\lambda_j}\right] \\
\overset{(i)}{\geq}& \sum_{j \in \mathcal{H}_0(t)} \mathbb{E}\left[\frac{\alpha_j}{dR_{j-1}^{\mathrm{d}}+1}\right] = \mathbb{E}\left[\text{mem-FDP}^*(t)\right],
\end{aligned}
$$

where inequality (i) holds because $\mathbb{E}\left[\mathbb{1}\left\{e_j < \frac{1}{\lambda_j}\right\} \mid \mathcal{F}_{j-1}\right] \geq 1-\lambda_j$ by the property of e-values. By Theorem 4.1, we have $\text{mem-FDR}(t) \leq \alpha$ for all $t$. $\square$

## A.6. Proof of Theorem 4.3

*Proof.* Recall that the decision $\delta_t = \mathbb{1}\{p_t \leq \alpha_t\}$ for each time $t$. Suppose a desired level $\alpha$ is given. For all time $t$, we have

$$
\begin{aligned}
\text{FDR}(t) =& \mathbb{E}\left[\frac{\sum_{j \in \mathcal{H}_0(t)} \delta_j}{R_t \vee 1}\right] \overset{(i)}{\leq} \mathbb{E}\left[\sum_{j \in \mathcal{H}_0(t)} \frac{\delta_j}{R_{j-1}+1}\right] = \mathbb{E}\left[\sum_{j \in \mathcal{H}_0(t)} \frac{\mathbb{1}\{p_j \leq \alpha_j\}}{R_{j-1}+1}\right] \\
=& \mathbb{E}\left[\sum_{j \in \mathcal{H}_0(t)} \frac{\mathbb{E}\left[\mathbb{1}\{p_j \leq \alpha_j\} \mid \mathcal{F}_{j-1}\right]}{R_{j-1}+1}\right] \overset{(ii)}{\leq} \mathbb{E}\left[\sum_{j \in \mathcal{H}_0(t)} \frac{\alpha_j}{R_{j-1}+1}\right] = \mathbb{E}\left[\text{FDP}_{\mathrm{e}}^*(t)\right] \overset{(iii)}{\leq} \alpha,
\end{aligned}
$$

where the inequality (i) holds since $R_{j-1} + 1 \leq (R_t \vee 1)$ for every $j \in \{j \leq t : \delta_j = 1\}$ by definition, the inequality (ii) follows after taking iterated expectations by conditioning on $\mathcal{F}_{j-1}$ and then applying the conditionally super-uniform property of p-values, and inequality (iii) holds by condition. Thus, we finish the whole proof. $\square$

## B. Deferred Discussions

### B.1. Definition of PRDS

Fisher (2024) introduced an online version of the well-known positive regression dependence on a subset (PRDS) condition proposed by Benjamini & Yekutieli (2001), considered the performance of LORD++ and SAFFRON, and proved online FDR control when the testing statistics, p-values, are conditional PRDS.

Before formally defining PRDS, it is necessary to first introduce the concept of increasing sets. A set $I \in \mathbb{R}^K$ is called increasing if $\mathbf{x} \in I$ implies $\mathbf{y} \in I$ for all $\mathbf{y} \geq \mathbf{x}$. Here $\mathbf{y} \geq \mathbf{x}$ implies that each component of $\mathbf{y}$ is no smaller than the corresponding component of $\mathbf{x}$.

**Definition B.1.** (Conditional PRDS between p-values; Fisher, 2024) The p-values are conditional PRDS if for each time $t$, any $j \leq t$ satisfying $j \in \mathcal{H}_0(t)$, and increasing set $I \subset \mathbb{R}^t$, the probability $\mathbb{P}((p_1, \ldots, p_t) \in I \mid p_j = u, \mathcal{F}_{j-1})$ is non-decreasing in $u$.

Wang & Ramdas (2022) discussed the PRDS condition on the studies of e-values in the offline setting. We generalize it to the version applicable to the online scenario. A set $D \in \mathbb{R}^K$ is called decreasing if $\mathbf{x} \in I$ implies $\mathbf{y} \in I$ for all $\mathbf{y} \leq \mathbf{x}$.

**Definition B.2.** (Conditional PRDS between e-values) The e-values are conditional PRDS if for each time $t$, any $j \leq t$ satisfying $j \in \mathcal{H}_0(t)$, and decreasing set $D \subset \mathbb{R}^t$, the probability $\mathbb{P}((e_1, \ldots, e_t) \in D \mid e_j = u, \mathcal{F}_{j-1})$ is non-increasing in $u$.

### B.2. Choices for $\omega_1$

In this section, we further elaborate on the motivation and advantage of updating $\omega_t$ in (9) from the RAI perspective. Building on this, we provide a theoretical justification for the recommended choice of $\omega_1 = O(1/T)$ and present supporting experimental results.

In the e-LORD and e-SAFFRON algorithms, $\omega_t$ controls the proportion of the remaining $\alpha$-wealth allocated to the current testing. A larger $\omega_t$ indicates that more $\alpha$-wealth is currently invested, which also implies a greater possibility of rejecting the current hypothesis, and meanwhile, it will be more possible to exhaust the entire wealth. Therefore, we prioritize updating $\omega_t$ from the RAI perspective, dynamically allocating the testing levels and enabling data-driven updates to achieve higher power. In contrast, the testing levels $\alpha_t$ in e-LOND are derived from a pre-specified decay sequence that sums to 1 (Xu & Ramdas, 2024).

A simplified version of (9) is to set $\varphi = \psi = 0$ and thus $\omega_t = \omega_1$ for all $t$. In this case, a natural and reasonable way to choose $\omega_1$ is to assign equal weight at each time point, i.e., $\omega_1 = 1/T$, motivating the choice of initial value for dynamic updates.

Empirical results support this analysis. The power results for different choices of $\omega_1$ across varying $T$ under an AR(1) model, introduced in Appendix C.2, are shown in Table 3. From Table 3, our algorithms with $\omega_1 = 1/T$ achieve the highest power and have the latest time of the last rejection and the largest tail testing level, supporting potential subsequent long-term testing. Moreover, it can be seen that the updates in e-LORD and e-SAFFRON are data-driven: as $T$ varies, the remaining wealth for these algorithms does not change significantly and shows robustness. In contrast, e-LOND uses a pre-specified allocation ratio, and as $T$ increases, the $\alpha$-wealth at time $T$ of e-LOND diminishes progressively.

### B.3. Recursive Update Forms of e-LORD & e-SAFFRON

In this section, we provide recursive update forms of e-LORD and e-SAFFRON, respectively. The computation is highly efficient and memory-friendly since the update of both $\omega_t$ (expressed in a recursive form in (9)) and $\alpha_t$ can be expressed in a recursive form as follows.

Table 3. Average results under an $\mathrm{AR}(1)$ model over 100 repetitions with $\mu_c = 4$, $\pi_1 = 0.4$, and $\alpha = 0.05$.

| $T$ | Method | $\omega_1$ | Power (%) | Time of the last rejection | $\alpha_T/\alpha(\times 10^{-4})$ |
|---|---|---|---|---|---|
| | e-LORD | $1/T$ | 70.0 | 498 | 1031.0 |
| | | $1/\sqrt{T}$ | 22.2 | 275 | 0.0 |
| | | $1/T^2$ | 8.6 | 483 | 0.7 |
| 500 | e-SAFFRON | $1/T$ | 70.5 | 498 | 1368.1 |
| | | $1/\sqrt{T}$ | 38.7 | 441 | 0.0 |
| | | $1/T^2$ | 8.0 | 483 | 0.6 |
| | e-LOND | – | 30.9 | 491 | 2.5 |
| | e-LORD | $1/T$ | 70.1 | 998 | 1029.0 |
| | | $1/\sqrt{T}$ | 16.2 | 406 | 0.0 |
| | | $1/T^2$ | 4.5 | 962 | 0.2 |
| 1000 | e-SAFFRON | $1/T$ | 70.9 | 998 | 1366.7 |
| | | $1/\sqrt{T}$ | 28.0 | 706 | 0.0 |
| | | $1/T^2$ | 4.2 | 958 | 0.2 |
| | e-LOND | – | 23.9 | 983 | 1.0 |

To compute $\alpha_t$ of e-LORD in Algorithm 1, we define the remaining wealth as

$$\mathrm{rw}_t^{\text{e-LORD}} = \alpha - \sum_{j=1}^{t-1} \frac{\alpha_j}{R_{j-1}+1}$$

and update

$$\alpha_t = \omega_t \, \mathrm{rw}_t^{\text{e-LORD}} (R_{t-1}+1).$$

A similar recursive form of $\alpha_t$ of e-SAFFRON in Algorithm 2 can be obtained by defining the remaining wealth as

$$\mathrm{rw}_t^{\text{e-SAFFRON}} = \alpha(1-\lambda) - \sum_{j=1}^{t-1} \frac{\alpha_j \mathbb{1}\{e_j < 1/\lambda\}}{R_{j-1}+1}$$

and update

$$\alpha_t = \omega_t \, \mathrm{rw}_t^{\text{e-SAFFRON}} (R_{t-1}+1).$$

Through these formulations, the update of testing levels $\alpha_t$ is expressed as a recursive relationship based on information from the previous time step, allowing us to compute it recursively and efficiently.

We evaluate the runtime of various algorithms in the experiments under an $\mathrm{AR}(1)$ model, introduced in Appendix C.2, and the results are included in Table 4. It shows that the e-LORD and e-SAFFRON algorithms are computationally efficient.

Table 4. Average runtime of different algorithms under an $\mathrm{AR}(1)$ model over 100 repetitions.

| | e-LORD | e-SAFFRON | e-LOND | LORD++ | SAFFRON | SupLORD |
|---|---|---|---|---|---|---|
| Runtime ($\times 10^{-4}$s) | 9.7 | 18.8 | 18.0 | 9.9 | 8.1 | 72.1 |

### B.4. Extension of the e-GAI Framework to p-values

In this section, we adapt the e-GAI framework to p-values and analyze the corresponding algorithms for the long-term performance following the same strategy in Section 3 and Section 4.1. Recall that the decision $\delta_t = \mathbb{1}\{p_t \le \alpha_t\}$ for each time $t$.

**Extension of e-LORD & e-SAFFRON to p-values.** Theorem 4.3 in the main text provides a general theoretical result for FDR control with conditionally super-uniform p-values. Leveraging Theorem 4.3, both e-LORD and e-SAFFRON can be adapted to the corresponding version applicable to p-values by replacing $\mathbb{1}\{e_t \geq 1/\alpha_t\}$ with $\mathbb{1}\{p_t \leq \alpha_t\}$. Specifically, $\widehat{\mathrm{FDP}}_{\mathrm{e}}^{\mathrm{LORD}}(t)$ in (7) and the update rule for $\alpha_t$ in (8) of the e-LORD algorithm can be directly adapted to p-values. The e-SAFFRON algorithm estimates the proportion of true nulls, requiring a slight modification when converting e-values to p-values. Considering $\lambda_t \equiv \lambda$, the modified FDP overestimate and testing levels of e-SAFFRON are respectively given by $\sum_{j=1}^{t} \frac{\alpha_j}{R_{j-1}+1} \frac{\mathbb{1}\{p_j>\lambda\}}{1-\lambda}$ and $\alpha_t = \omega_t \left( \alpha(1-\lambda) - \sum_{j=1}^{t-1} \frac{\alpha_j \mathbb{1}\{p_j>\lambda\}}{R_{j-1}+1} \right) (R_{t-1}+1)$.

To clearly distinguish the algorithms, we refer to the versions of e-LORD and e-SAFFRON adapted to p-values as pL-RAI and pS-RAI, respectively, emphasizing that the testing statistics are p-values and testing levels $\alpha_t$ are updated by the RAI strategy. Both pL-RAI and pS-RAI can realize provable FDR control under conditional super-uniformity. To avoid notational confusion, we define $\widehat{\mathrm{FDP}}^{\mathrm{pL\text{-}RAI}}(t) := \widehat{\mathrm{FDP}}_{\mathrm{e}}^{\mathrm{LORD}}(t) = \sum_{j=1}^{t} \frac{\alpha_j}{R_{j-1}+1}$ and $\widehat{\mathrm{FDP}}^{\mathrm{pS\text{-}RAI}}(t) := \sum_{j=1}^{t} \frac{\alpha_j}{R_{j-1}+1} \frac{\mathbb{1}\{p_j>\lambda\}}{1-\lambda}$.

**Proposition B.3.** *Suppose the online p-values are conditionally super-uniform in* (1).

(a) *For $\alpha_t \in \mathcal{F}_{t-1}$ satisfying $\widehat{\mathrm{FDP}}^{\mathrm{pL\text{-}RAI}}(t) \leq \alpha$, we have $\mathrm{FDR}(t) \leq \alpha$ for all t.*

(b) *Given a predictable sequence $\{\lambda_t\}_{t=1}^{\infty}$, for $\alpha_t \in \mathcal{F}_{t-1}$ satisfying $\widehat{\mathrm{FDP}}^{\mathrm{pS\text{-}RAI}}(t) \leq \alpha$, we have: (i) $\mathbb{E}\left[ \widehat{\mathrm{FDP}}^{\mathrm{pS\text{-}RAI}}(t) \right] \geq \mathbb{E}\left[ \mathrm{FDP}_{\mathrm{e}}^*(t) \right]$ where $\mathrm{FDP}_{\mathrm{e}}^*(t)$ is defined in (6), and (ii) $\mathrm{FDR}(t) \leq \alpha$ for all t.*

The proof strategy for Proposition B.3 follows the same approach as Proposition 3.2 and Proposition 3.4, with the key distinction residing in the application of the inequality $\mathbb{E}\left[\mathbb{1}\{p_j > \lambda_j\} \mid \mathcal{F}_{j-1}\right] \geq 1 - \lambda_j$, which is derived from the conditionally super-uniform property of p-values. Hence, we omit the detailed derivations.

**Long-Term Performance of pL-RAI & pS-RAI.** To control mem-FDR using p-values, we demonstrate that $\mathrm{mem\text{-}FDP}^*(t)$ in (11) can still serve as an oracle estimate of mem-FDR. By controlling this estimator to be bounded, we can achieve mem-FDR control for p-values satisfying the conditionally super-uniform property in (1).

**Theorem B.4.** *Suppose the online p-values are conditionally super-uniform in* (1). *If $\mathrm{mem\text{-}FDP}^*(t)$ in (11) satisfies $\mathbb{E}\left[\mathrm{mem\text{-}FDP}^*(t)\right] \leq \alpha$, then $\mathrm{mem\text{-}FDR}(t) \leq \alpha$ for all t.*

Leveraging Theorem B.4, both mem-e-LORD and mem-e-SAFFRON can be adapted to the corresponding version applicable to p-values. Specifically, $\mathrm{mem\text{-}}\widehat{\mathrm{FDP}}^{\mathrm{LORD}}(t)$ in (12) and the update rule for $\alpha_t$ in (13) of the mem-e-LORD algorithm can be directly adapted to p-values. Considering $\lambda_t \equiv \lambda$, the modified mem-FDP overestimate and testing levels of mem-e-SAFFRON are respectively given by $\sum_{j=1}^{t} \frac{\alpha_j}{dR_{j-1}^{\mathrm{d}}+1} \frac{\mathbb{1}\{p_j>\lambda\}}{1-\lambda}$ and $\alpha_t = \omega_t \left( \alpha(1-\lambda) - \sum_{j=1}^{t-1} \frac{\alpha_j \mathbb{1}\{p_j>\lambda\}}{dR_{j-1}^{\mathrm{d}}+1} \right) (dR_{t-1}^{\mathrm{d}}+1)$.

To clearly distinguish the algorithms, we refer to the versions of mem-e-LORD and mem-e-SAFFRON adapted to p-values as mem-pL-RAI and mem-pS-RAI, respectively. Both mem-pL-RAI and mem-pS-RAI can realize provable mem-FDR control under conditional super-uniformity. To avoid notational confusion, we define $\mathrm{mem\text{-}}\widehat{\mathrm{FDP}}^{\mathrm{pL\text{-}RAI}}(t) := \mathrm{mem\text{-}}\widehat{\mathrm{FDP}}^{\mathrm{LORD}}(t) = \sum_{j=1}^{t} \frac{\alpha_j}{dR_{j-1}^{\mathrm{d}}+1}$ and $\mathrm{mem\text{-}}\widehat{\mathrm{FDP}}^{\mathrm{pS\text{-}RAI}}(t) := \sum_{j=1}^{t} \frac{\alpha_j}{dR_{j-1}^{\mathrm{d}}+1} \frac{\mathbb{1}\{p_j>\lambda\}}{1-\lambda}$.

**Proposition B.5.** *Suppose the online p-values are conditionally super-uniform in* (1).

(a) *For $\alpha_t \in \mathcal{F}_{t-1}$ satisfying $\mathrm{mem\text{-}}\widehat{\mathrm{FDP}}^{\mathrm{pL\text{-}RAI}}(t) \leq \alpha$, we have $\mathrm{mem\text{-}FDR}(t) \leq \alpha$ for all t.*

(b) *Given a predictable sequence $\{\lambda_t\}_{t=1}^{\infty}$, for $\alpha_t \in \mathcal{F}_{t-1}$ satisfying $\mathrm{mem\text{-}}\widehat{\mathrm{FDP}}^{\mathrm{pS\text{-}RAI}}(t) \leq \alpha$, we have: (i) $\mathbb{E}\left[ \mathrm{mem\text{-}}\widehat{\mathrm{FDP}}^{\mathrm{pS\text{-}RAI}}(t) \right] \geq \mathbb{E}\left[\mathrm{mem\text{-}FDP}^*(t)\right]$ where $\mathrm{mem\text{-}FDP}^*(t)$ is defined in (11), and (ii) $\mathrm{mem\text{-}FDR}(t) \leq \alpha$ for all t.*

We omit the proofs of Theorem B.4 and Proposition B.5, as they follow analogous reasoning and use identical techniques to those previously demonstrated.

## B.5. Connection to Existing Methods in GAI under Independence

In this section, we demonstrate the FDR control with independent e-values, and compare the e-GAI framework with the GAI methods, highlighting that e-GAI is a unified framework.

When working with independent e-values, the correlation between the number of false rejections and the total number of rejections can be analyzed using the leave-one-out technique (Ramdas et al., 2018, Lemma 1), which ensures the control of FDR. This property is formalized as follows, paralleling Theorem 3.1 in the independent situation.

**Theorem B.6.** *Suppose online e-values are valid in* (2)*, and the null e-values are independent of each other and of the non-nulls. Under independence, let the oracle e-value-based estimate of FDP be given as*

$$\mathrm{FDP}^*_{\mathrm{e\text{-}ind}}(t) = \sum_{j\in\mathcal{H}_0(t)} \frac{\alpha_j}{R_t \vee 1}. \tag{14}$$

*If* $\mathbb{E}\left[\mathrm{FDP}^*_{\mathrm{e\text{-}ind}}(t)\right] \leq \alpha$ *and* $\{\alpha_t\}$ *is a monotone function of* $\delta_{t-1}$ *for all t, then* $\mathrm{FDR}(t) \leq \alpha$ *for all t.*

*Proof.* To control online FDR, we have

$$
\begin{aligned}
\mathrm{FDR}(t) =& \mathbb{E}\left[\frac{\sum_{j\in\mathcal{H}_0(t)} \delta_j}{R_t \vee 1}\right] = \mathbb{E}\left[\frac{\sum_{j\in\mathcal{H}_0(t)} \mathbb{1}\left\{e_j \geq \frac{1}{\alpha_j}\right\}}{R_t \vee 1}\right] = \mathbb{E}\left[\sum_{j\in\mathcal{H}_0(t)} \mathbb{E}\left[\frac{\mathbb{1}\left\{\frac{1}{e_j} \leq \alpha_j\right\}}{R_t \vee 1} \mid \mathcal{F}_{j-1}\right]\right] \\
&\overset{(i)}{\leq} \mathbb{E}\left[\sum_{j\in\mathcal{H}_0(t)} \mathbb{E}\left[\frac{\alpha_j}{R_t \vee 1} \mid \mathcal{F}_{j-1}\right]\right] = \mathbb{E}\left[\frac{\sum_{j\in\mathcal{H}_0(t)} \alpha_j}{R_t \vee 1}\right] \leq \alpha,
\end{aligned}
$$

where the inequality (i) uses the transformation from e-values into p-values and the leave-one-out technique as in the (Ramdas et al., 2017, Lemma 1) and (Ramdas et al., 2018, Lemma 1) due to the independence. Thus we finish the whole proof. $\square$

Theorem B.6 builds a bridge between our framework and previous methods based on p-values in prior works introduced in Section 2.2. Specifically, we established the connection between e-LORD and LORD, as well as between e-SAFFRON and SAFFRON, respectively.

**e-LORD & LORD++.** Within the e-GAI framework, we can overestimate $\mathrm{FDP}^*_{\mathrm{e\text{-}ind}}(t)$ in (14) by $\widehat{\mathrm{FDP}}^{\mathrm{LORD}}_{\mathrm{e\text{-}ind}}(t) := \frac{\sum_{j=1}^{t} \alpha_j}{R_t \vee 1}$, same as $\widehat{\mathrm{FDP}}^{\mathrm{LORD}}(t)$ in (4), in the independent scenario. Then the process of generating testing levels $\{\alpha_t\}$ by LORD++ (Ramdas et al., 2017) can be regarded as the e-LORD algorithm, as it satisfies the condition $\widehat{\mathrm{FDP}}^{\mathrm{LORD}}_{\mathrm{e\text{-}ind}}(t) \leq \alpha$. Therefore, LORD++ can be included within the e-GAI framework.

Furthermore, we can show the equivalence between e-LORD and LORD++. On the one hand, when the independent e-values $\{e_t\}$ are available at each time, then $1/e_t$ is a valid p-value by (23) in Appendix B.7, and applying LORD++ to $\{1/e_t\}$ is equivalent to applying e-LORD to $\{e_t\}$, wherein the testing levels $\{\alpha_t\}$ for e-LORD are derived following the LORD++ procedure (Ramdas et al., 2017), i.e.,

$$\alpha_t^{\mathrm{LORD++}} = \gamma_t W_0 + (\alpha - W_0)\gamma_{t-\tau_1} + \alpha \sum_{j:\tau_j < t, \tau_j \neq \tau_1} \gamma_{t-\tau_j}, \tag{15}$$

where $W_0 > 0$ is the initial $\alpha$-wealth, $\tau_t$ is the time of the $t$-th rejection and $\{\gamma_t\}$ is pre-specified non-negative sequence summing to one.

On the other hand, when the independent p-values $\{p_t\}$ are available instead, then $\mathbb{1}\{p_t \leq \alpha_t\}/\alpha_t$ is a valid e-value. This is because

$$\mathbb{E}\left[\frac{\mathbb{1}\{p_t \leq \alpha_t\}}{\alpha_t} \mid \mathcal{F}_{t-1}\right] \leq 1$$

due to the conditionally super-uniform property of $p_t$. Define $e_t = \mathbb{1}\{p_t \leq \alpha_t\}/\alpha_t$ and applying LORD++ to $\{p_t\}$ is equivalent to applying e-LORD to $\{e_t\}$ with the testing levels $\{\alpha_t\}$ generated in the same manner as LORD++ (Ramdas

et al., 2017). To prove it, denote $\delta_t^p = \mathbb{1}\{p_t \leq \alpha_t^p\}$ and $\delta_t^e = \mathbb{1}\{e_t \geq \frac{1}{\alpha_t^e}\}$ respectively to easily distinguish. It can be readily verified that $\delta_1^p = \delta_1^e$. Suppose for all $j \leq t-1$, we have $\delta_j^p = \delta_j^e$. Then for time $t$, $\alpha_t^p = \alpha_t^e$ are both generated by the same LORD++ algorithm in (15) and hence we omit the superscript. If $\delta_t^p = 1$, it follows that $p_t \leq \alpha_t$, which implies that $e_t = 1/\alpha_t$ and consequently $\delta_t^e = 1$. Conversely, if $\delta_t^p = 0$, it also holds that $\delta_t^e = 0$. The proof can then be concluded through recursive reasoning.

**e-SAFFRON & SAFFRON.** It is natural to derive the relationship between e-SAFFRON and SAFFRON. In the independent scenario, we can overestimate $\mathrm{FDP}_{\mathrm{e\text{-}ind}}^*(t)$ in (14) by $\widehat{\mathrm{FDP}}_{\mathrm{e\text{-}ind}}^{\mathrm{SAFFRON}}(t) := \frac{\sum_{j=1}^{t} \alpha_j \frac{\mathbb{1}\{e_j < 1/\lambda\}}{1-\lambda}}{R_t \vee 1}$, same as $\widehat{\mathrm{FDP}}^{\mathrm{SAFFRON}}(t)$ in (5) when $p_t = 1/e_t$. Therefore, the process of generating testing levels $\{\alpha_t\}$ by SAFFRON (Ramdas et al., 2018) can be regarded as the e-SAFFRON algorithm as it satisfies $\widehat{\mathrm{FDP}}_{\mathrm{e\text{-}ind}}^{\mathrm{SAFFRON}}(t) \leq \alpha$.

However, the conclusion on the equivalence between e-SAFFRON and SAFFRON differs from that between e-LORD and LORD++. When the independent e-values $\{e_t\}$ are available at each time, applying SAFFRON to valid p-values $\{p_t = 1/e_t\}$ is equivalent to applying e-SAFFRON to $\{e_t\}$, wherein the testing levels $\{\alpha_t\}$ for e-SAFFRON are derived following the SAFFRON procedure (Ramdas et al., 2018), i.e.,

$$\alpha_t^{\mathrm{SAFFRON}} = \min\left\{\lambda, W_0\gamma_{t-C_{0+}} + ((1-\lambda)\alpha - W_0)\gamma_{t-\tau_1-C_{1+}} + \sum_{j\geq 2}(1-\lambda)\alpha\gamma_{t-\tau_j-C_{j+}}\right\}. \tag{16}$$

Here $\lambda \in (0,1)$ is a user-chosen parameter and $C_{i+}(t) = \sum_{j=\tau_i+1}^{t-1} \mathbb{1}\{p_j \leq \lambda\}$.

The situation changes when the independent p-values are available. In addition to the conditions of independence, if the p-values further satisfy the conditionally uniformly distributed in (1), then there are no valid e-values in (2) such that applying SAFFRON to $\{p_t\}$ is equivalent to applying e-SAFFRON to those e-values, wherein the testing levels $\{\alpha_t\}$ for e-SAFFRON are derived following the SAFFRON procedure (Ramdas et al., 2018). We demonstrate it by contradiction. Suppose there exists valid e-values $\{e_t\}$ with $e_t = f^c(p_t)$ such that applying SAFFRON to $\{p_t\}$ is equivalent to applying e-SAFFRON to $\{e_t\}$, wherein the testing levels $\{\alpha_t\}$ for e-SAFFRON are derived following the SAFFRON procedure in (16). Since the p-values are conditionally uniformly distributed, the p-values are also continuously distributed. Note that the p-values play three different roles in the SAFFRON algorithm: rejection if $p_t \leq \alpha_t$, candidate if $\alpha < p_t \leq \lambda$, and inclusion in the estimate of $\mathrm{FDP}^*$ if $p_t > \lambda$. Correspondingly, the e-values $f^c(p_t)$ satisfy $f^c(p_t) \geq 1/\alpha_t$ if $p_t \leq \alpha_t$, $f^c(p_t) \geq 1/\lambda$ if $\alpha < p_t \leq \lambda$, and $f^c(p_t) < 1/\lambda$ if $p_t > \lambda$. By the above assumptions, we have

$$\mathbb{E}\left[f^c(p_t)\right] = \int f^c(p)\mathbb{1}\{p \leq \alpha_t\}dp + \int f^c(p)\mathbb{1}\{\alpha_t < p \leq \lambda\}dp + \int f^c(p)\mathbb{1}\{p > \lambda\}dp$$

$$\geq \int \frac{1}{\alpha_t}\mathbb{1}\{p \leq \alpha_t\}dp + \int \frac{1}{\lambda}\mathbb{1}\{\alpha_t < p \leq \lambda\}dp + \int f^c(p)\mathbb{1}\{p > \lambda\}dp$$

$$\geq 1 + \int \frac{1}{\lambda}\mathbb{1}\{\alpha_t < p \leq \lambda\}dp$$

$$> 1$$

with $\alpha_t$ generated by SAFFRON, which contradicts the assumption that $f^c(p_t)$ is a valid e-value. In summary, there are no valid e-values transformed by p-values that also serve the three roles accordingly in e-SAFFRON. This conclusion implies that there may be a loss of power when testing by e-values compared to using p-values in the independent case.

### B.6. Alternative Expressions of e-LORD & e-SAFFRON

In this section, we provide alternative expressions of e-LORD and e-SAFFRON, respectively. These expressions may aid in better understanding the dynamics of the allocation process and how the e-LOND algorithm can be considered as a special case within the e-GAI framework with design $\omega_t$ from a given sequence $\{\gamma_t\}$.

**e-LORD.** Recall the e-LORD algorithm updates testing levels as $\alpha_1 = \alpha\omega_1$ and for $t \geq 2$:

$$\alpha_t = \omega_t\left(\alpha - \sum_{j=1}^{t-1}\frac{\alpha_j}{R_{j-1}+1}\right)(R_{t-1}+1), \tag{17}$$

where $\omega_1, \ldots, \omega_t \in \mathcal{F}_{t-1}$ are the allocation coefficients, updated as in (9). By recursion and calculation, (17) can be simplified to

$$\alpha_t = \alpha(R_{t-1} + 1)\omega_t \prod_{j=1}^{t-1}(1 - \omega_j). \tag{18}$$

**e-SAFFRON.** We follow the same routine as above. Recall the e-SAFFRON algorithm updates testing levels as $\alpha_1 = \alpha(1 - \lambda)\omega_1$ and for $t \geq 2$:

$$\alpha_t = \omega_t \left( \alpha(1 - \lambda) - \sum_{j=1}^{t-1} \frac{\alpha_j \mathbb{1}\left\{e_j < \frac{1}{\lambda}\right\}}{R_{j-1} + 1} \right)(R_{t-1} + 1), \tag{19}$$

where $\omega_1, \ldots, \omega_t \in \mathcal{F}_{t-1}$ are the allocation coefficients, updated as in (9). By recursion and calculation, (19) can be simplified to

$$\alpha_t = \alpha(1 - \lambda)(R_{t-1} + 1)\omega_t \prod_{j=1}^{t-1}(1 - \omega_j \mathbb{1}\{e_j < 1/\lambda\}). \tag{20}$$

(18) and (20) provide an alternative perspective for understanding the allocation process. In addition to the item regarding the number of previous rejections, the remaining items focus on the process of wealth allocation, in which the total wealth $\alpha$ is dynamically allocated by $\omega_t$ and the prior allocation coefficients $\{\omega_j\}_{j=1}^{t-1}$.

### B.7. Construction of Online e-values

In this section, we offer useful suggestions about the construction of valid e-values that can be applied to the online setting. Recall that a non-negative variable $e_t$ is a valid online e-value if it satisfies the conditional validity property:

$$\mathbb{E}[e_t \mid \mathcal{F}_{t-1}] \leq 1 \text{ if } \theta_t = 0.$$

**Likelihood ratio e-values.** Assume that the null conditional distribution $F_{0,t|\mathcal{F}_{t-1}}$ is known for all $t$ and denote the corresponding density function $f_{0,t|\mathcal{F}_{t-1}}$. At each time $t$, the conditional distribution $F_{t|\mathcal{F}_{t-1}}$ and the corresponding density $f_{t|\mathcal{F}_{t-1}}$ are unknown, and we can estimate $f_{t|\mathcal{F}_{t-1}}$ using parametric or nonparametric methods. Then we can construct the likelihood ratio e-value as

$$e_t = \frac{\hat{f}_{t|\mathcal{F}_{t-1}}(x_t)}{f_{0|\mathcal{F}_{t-1}}(x_t)} \tag{21}$$

for each time $t$. The e-value $e_t$ in (21) is valid since

$$\mathbb{E}\left[e_t \mid \mathcal{F}_{t-1}\right] = \int \frac{\hat{f}_{t|\mathcal{F}_{t-1}}(x_t)}{f_{0|\mathcal{F}_{t-1}}(x_t)} f_{0|\mathcal{F}_{t-1}}(x_t)dx_t = \int \hat{f}_{t|\mathcal{F}_{t-1}}(x_t)dx_t = 1.$$

**p-value-based e-values.** In case the p-values associated with each hypothesis are available, it is possible to convert them to e-values using a 'p-to-e calibrator' (Shafer et al., 2011), albeit with possible power loss (Vovk & Wang, 2021). A 'p-to-e calibrator' is a decreasing function $f^c : [0, 1] \mapsto [0, \infty]$, such that $\int_0^1 f^c(s)ds = 1$. Then we can construct the p-value-based e-value as

$$e_t = f^c(p_t) \tag{22}$$

for each time $t$, where $p_t$ is the corresponding p-value. The e-value $e_t$ in (22) is valid as long as the p-value $p_t$ is conditionally super-uniformly distributed under the null. Note that the choices for $f^c$ vary. For example, we can simply take $f^c(s) = \eta s^{\eta-1}$ for some $\eta \in (0, 1)$ (Shafer, 2021; Vovk & Wang, 2021).

Note that an e-value $e_t$ can be naturally transformed into a p-value $p_t$ by $p_t = \min\{1/e_t, 1\}$. This is because

$$\mathbb{P}(p_t \leq u \mid \mathcal{F}_{t-1}) \leq \mathbb{P}(e_t \geq 1/u \mid \mathcal{F}_{t-1}) \leq u\mathbb{E}[e_t \mid \mathcal{F}_{t-1}] \leq u \tag{23}$$

for all $u \in (0, 1)$ by Markov's inequality.

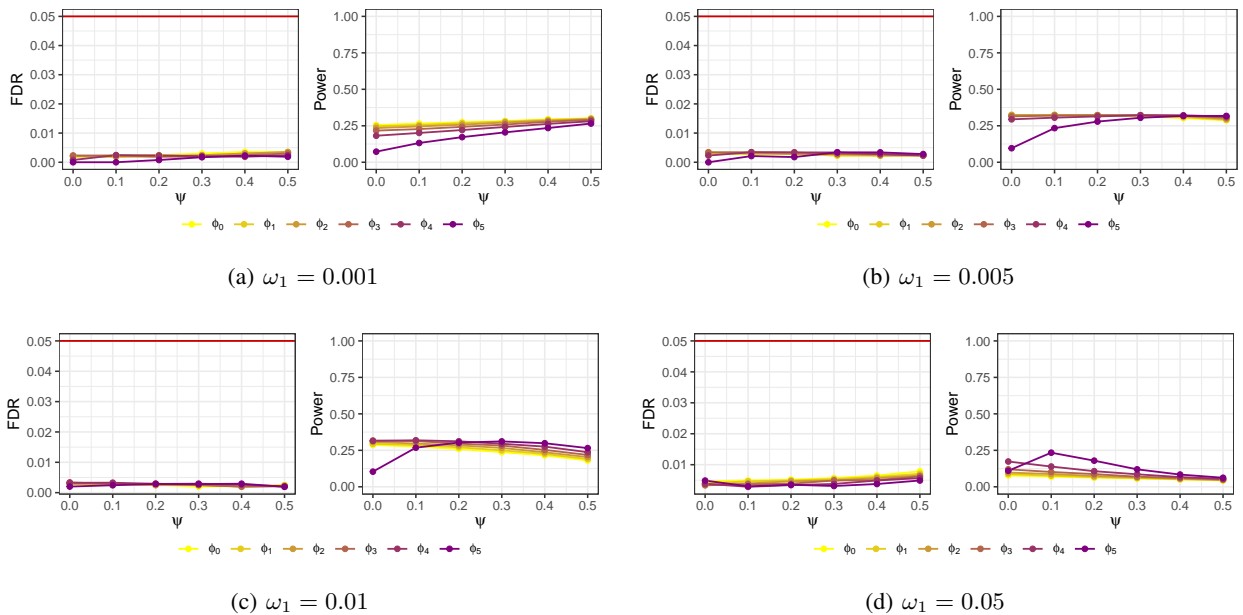

(a) $\omega_1 = 0.001$

(b) $\omega_1 = 0.005$

(c) $\omega_1 = 0.01$

(d) $\omega_1 = 0.05$

*Figure 4.* Empirical FDR and power with standard error versus $\psi$ for e-LORD, with $\rho = 0.5$, $L = 30$, $\mu_\mathrm{c} = 3$ and $\pi_1 = 0.2$. The $\varphi_0, \varphi_1, \ldots, \varphi_5$ methods correspond to $\varphi = 0, 0.1, \ldots, 0.5$, respectively. The parameter configuration $(\omega_1 = 0.005, \phi = 0.5, \varphi = 0.5)$ yielded the most favorable performance.

## C. Additional Simulation Results

### C.1. Simulation: FDR control

In this section, we investigate the impact of different parameters on the performance of our proposed method and, based on our findings, provide recommendations for appropriate parameter choices. Simulation results under other settings are also present to show the performance of e-LORD, e-SAFFRON, pL-RAI, pS-RAI and other algorithms.

**Updating $\omega_t$ with different parameters $\omega_1, \varphi, \psi$.**  To investigate the performance of e-GAI with varying parameters $\omega_1, \varphi, \psi$, we conduct a series of experiments on e-LORD, e-SAFFRON ($\lambda = 0.1$), pL-RAI and pS-RAI ($\lambda = 0.1$), following the settings outlined in Section 5.1, with $\alpha = 0.05$, $\rho = 0.5$, $L = 30$, $\mu_\mathrm{c} = 3$, and $\pi_1 = 0.2$.

We vary $\omega_1$, $\varphi$, and $\psi$, comparing the performance of the e-LORD, e-SAFFRON, pL-RAI, and pS-RAI under various parameter settings. Figures 4 to 7 show the results of e-LORD, e-SAFFRON, pL-RAI, and pS-RAI with different parameters, respectively. As shown in Figures 4 to 7, the e-LORD, e-SAFFRON, pL-RAI, and pS-RAI algorithms successfully achieve FDR control across all settings, which aligns with the theoretical guarantees. The power of the algorithm is affected by variations in the parameters $\omega_1, \varphi, \psi$. When $\omega_1 = 0.005$, $\varphi = \psi = 0.05$, all of e-LORD, e-SAFFRON, pL-RAI, and pS-RAI demonstrate high statistical power. Therefore, the parameter settings are chosen as $\omega_1 = 0.005$, $\varphi = \psi = 0.05$ in our experiments.

**Using different constant parameter $\lambda$ in e-SAFFRON & pS-RAI.**  To evaluate the performance of e-SAFFRON and pS-RAI with different values of $\lambda$, we adopt the same experimental settings as described in Section 5.1.

Figure 8 show the results of e-SAFFRON and pS-RAI with different $\lambda$. As illustrated in Figure 8, both e-SAFFRON and pS-RAI maintain FDR control across all different values of $\lambda$, which is consistent with the theoretical guarantees. It is evident that when $\lambda = 0.1$, both the e-SAFFRON and pS-RAI algorithms attain the highest statistical power. Therefore, we recommend setting $\lambda = 0.1$ as the default value when applying the e-SAFFRON and pS-RAI algorithms.

**Setting different correlation parameter $\rho$.**  To assess the performance of e-LORD, e-SAFFRON, pL-RAI and pS-RAI under varying levels of dependence, we adopt the same experimental settings as described in Section 5.1, and compare them

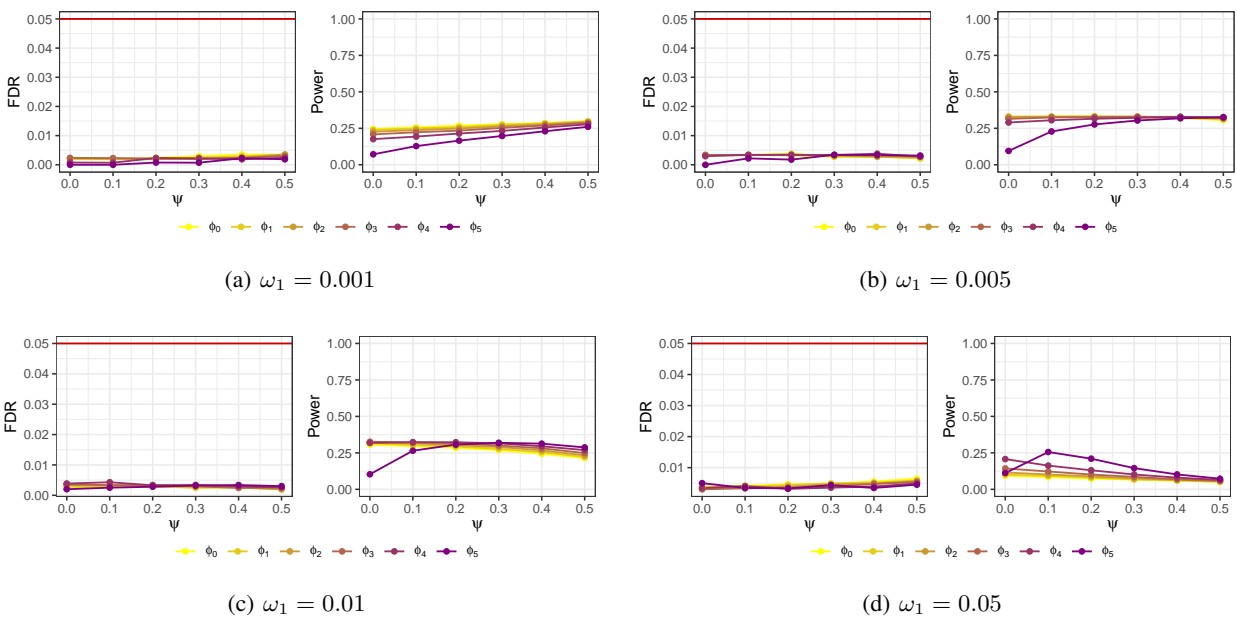

*Figure 5.* Empirical FDR and power with standard error versus $\psi$ for e-SAFFRON, with $\lambda = 0.1$, $\rho = 0.5$, $L = 30$, $\mu_c = 3$ and $\pi_1 = 0.2$. The $\varphi_0, \varphi_1, \ldots, \varphi_5$ methods correspond to $\varphi = 0, 0.1, \ldots, 0.5$, respectively. The parameter configuration ($\omega_1 = 0.005, \phi = 0.5, \varphi = 0.5$) yielded the most favorable performance.

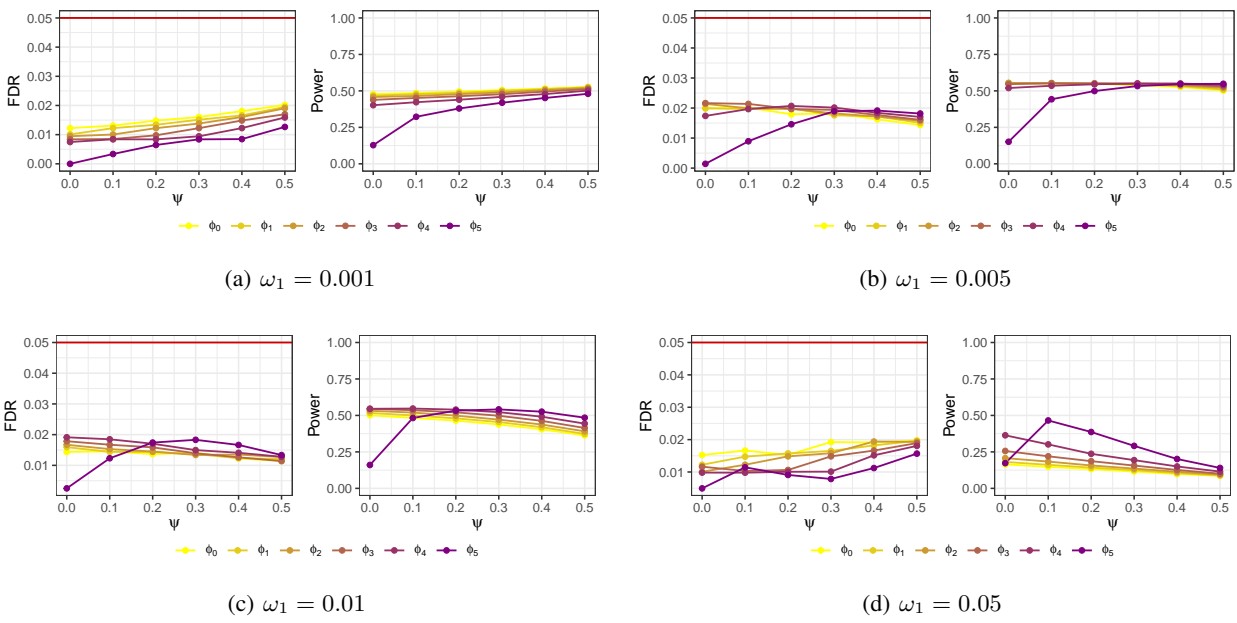

*Figure 6.* Empirical FDR and power with standard error versus $\psi$ for pL-RAI, with $\rho = 0.5$, $L = 30$, $\mu_c = 3$ and $\pi_1 = 0.2$. The $\varphi_0, \varphi_1, \ldots, \varphi_5$ methods correspond to $\varphi = 0, 0.1, \ldots, 0.5$, respectively. The parameter configuration ($\omega_1 = 0.005, \phi = 0.5, \varphi = 0.5$) yielded the most favorable performance.

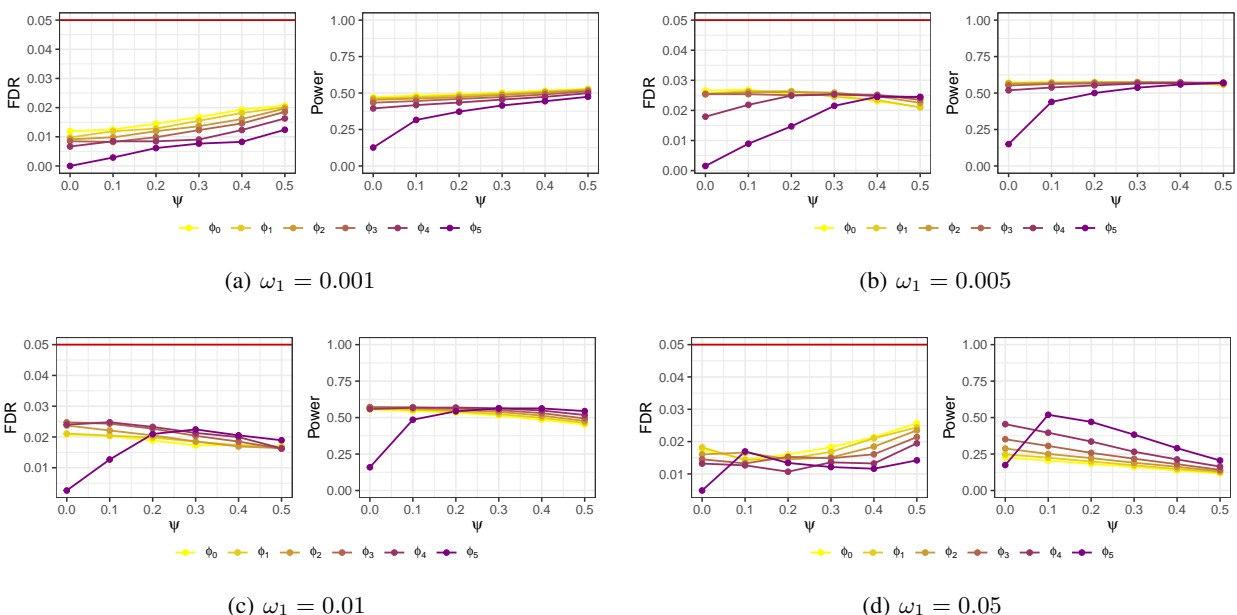

*Figure 7.* Empirical FDR and power with standard error versus $\psi$ for pS-RAI, with $\lambda = 0.1$, $\rho = 0.5$, $L = 30$, $\mu_c = 3$ and $\pi_1 = 0.2$. The $\varphi_0, \varphi_1, \ldots, \varphi_5$ methods correspond to $\varphi = 0, 0.1, \ldots, 0.5$, respectively. The parameter configuration $(\omega_1 = 0.005, \phi = 0.5, \varphi = 0.5)$ yielded the most favorable performance.

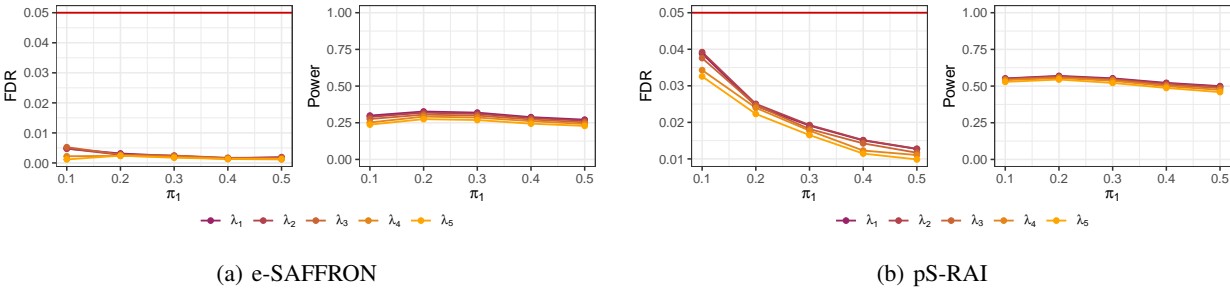

*Figure 8.* Empirical FDR and power with standard error versus proportion of alternative hypotheses $\pi_1$ for e-SAFFRON and pS-RAI, with $\rho = 0.5$ and $\mu_c = 3$. The $\lambda_1, \lambda_2, \ldots, \lambda_5$ methods correspond to $\lambda = 0.1, 0.2, \ldots, 0.5$, respectively. The parameter $\lambda = 0.1$ yielded the most favorable performance.

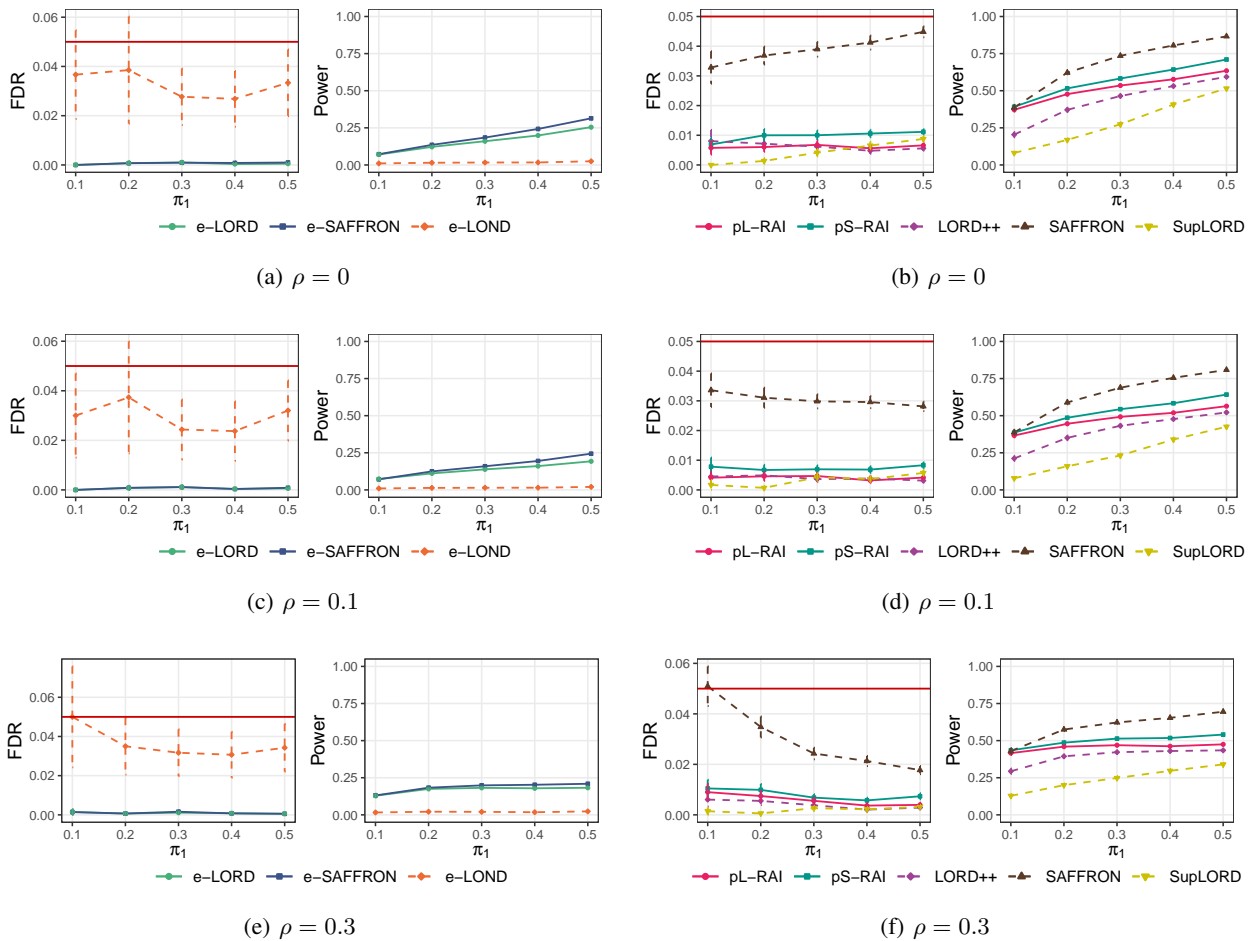

*Figure 9.* Empirical FDR and power with standard error versus proportion of alternative hypotheses $\pi_1$ for e-LORD, e-SAFFRON, e-LOND, pL-RAI, pS-RAI, LORD++, SAFFRON and SupLORD, with varying $\rho$ and fixed $\mu_c = 3$. Our pL-RAI and pS-RAI consistently outperform LORD++ and SupLORD, whereas SAFFRON fails to control FDR when $\rho$ is large and $\pi_1$ is small.

with e-LOND, LORD++, SAFFRON and SupLORD.

Figure 9 show the results of various methods with varying $\rho$ when $\mu_c = 3$. The performance of each method is similar to that in the main body. For e-value-based methods, irrespective of the correlation parameter $\rho$, e-LORD, e-SAFFRON, and e-LOND successfully achieve FDR control. Moreover, e-LORD and e-SAFFRON exhibit substantially higher statistical power than e-LOND across all settings by dynamically updating the testing levels, leading to more effective discoveries. For p-value-based methods, as shown in Figure 9, SAFFRON exhibits FDR inflation under such conditions. Besides, our pL-RAI and pS-RAI always gain higher power than LORD++ and SupLORD. In the independence case ($\rho = 0$), each method successfully maintains FDR control empirically, as guaranteed by their theoretical results.

### C.2. Simulation: mem-FDR control

We consider a new experimental setup where the samples follow a time-varying auto-regressive AR(1) model. For each time $t$, $X_t = \rho_t X_{t-1} + \mu_t + \varepsilon_t$ with $\varepsilon_t \stackrel{\text{i.i.d.}}{\sim} \mathcal{N}(0,1)$, where the auto-regressive coefficient $\rho_t = \frac{2}{1+\exp\left(-\eta(t-t_0)\right)} - 1 \in (-1,1)$. We set $\eta = 0.01$ and $t_0 = T/2$. We aim to test whether there is a positive drift and the null hypothesis takes $\mathbb{H}_t : \mu_t = 0$ for each time $t \in [T]$. The true labels $\theta_t$ is generated from $\text{Bernoulli}(\pi_1)$ and the positive drift $\mu_t = \mu_c > 0$ if $\theta_t = 1$. Note that the correlation coefficient $\rho_t$ between adjacent samples varies over time, resulting in a complex correlation structure.

We compare the performance of mem-e-LORD, mem-e-SAFFRON, mem-pL-RAI and mem-pS-RAI with mem-LORD++

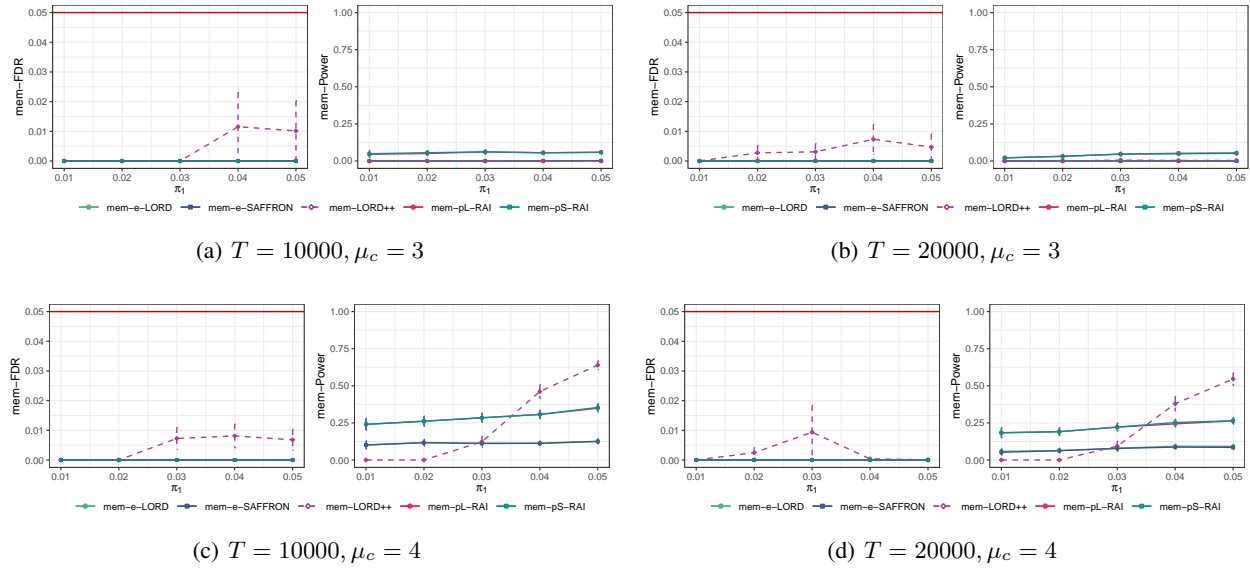

*Figure 10.* Empirical mem-FDR and mem-Power with standard error versus proportion of alternative hypotheses $\pi_1$ for mem-e-LORD, mem-e-SAFFRON, mem-LORD++, mem-pL-RAI and mem-pS-RAI with $\mu_c = 3, 4$, at $T = 10000, 20000$. All procedures maintain FDR control. Our e-GAI and RAI methods exhibit superior mem-power than mem-LORD++ when $\mu_c$ or $\pi_1$ is small.

at $T \in \{10000, 20000\}$ with different proportions of alternative hypotheses $\pi_1$ and strengths of signals $\mu_c$. We utilize the AR(1) model and the normal distribution of noise to calculate e-values and p-values that satisfy (2) and (1), respectively.

As shown in Figure 10, all methods successfully control the mem-FDR. Our procedures perform well over long-term testing periods, especially with sparse alternatives. In such cases, these algorithms achieve higher mem-Power than mem-LORD++. The mem-Power refers to the decaying memory power in (Ramdas et al., 2017), defined as

$$\text{mem-Power}(t) := \mathbb{E}\left[ \frac{\sum_{j \notin \mathcal{H}_0(t)} d^{t-j} \delta_j}{\sum_{j \notin \mathcal{H}_0(t)} d^{t-j}} \right].$$

### C.3. More results for Real Data: NYC Taxi Anomaly Detection

We adjust $\alpha = 0.2$ and apply pL-RAI, pS-RAI, SAFFRON, LORD++, and SupLORD here to analyze this dataset. We construct two-sided Gaussian p-values by estimating the mean and variance from the residuals.

We compare their performance in terms of the proportion of discoveries out of marked anomalous regions, denoted here as $\widehat{\text{FDP}}$ and the number of discovered anomalous regions in Table 5. We observe that the $\widehat{\text{FDP}}$ of LORD++ and SAFFRON far exceeds the testing level $\alpha = 0.2$ and SupLORD slightly exceeds $\alpha$. Meanwhile, pL-RAI and pS-RAI effectively maintain $\widehat{\text{FDP}}$ below the target level. As illustrated in Figure 11, pL-RAI and pS-RAI identify many points within the anomalous regions.

*Table 5.* Proportion of points rejected out of anomalous regions.

| Method | pL-RAI | pS-RAI | LORD++ | SAFFRON | SupLORD |
|---|---|---|---|---|---|
| $\widehat{\text{FDP}}$ | 0.197 | 0.195 | 0.261 | 0.361 | 0.217 |
| Num Discovery | 201 | 259 | 257 | 595 | 406 |

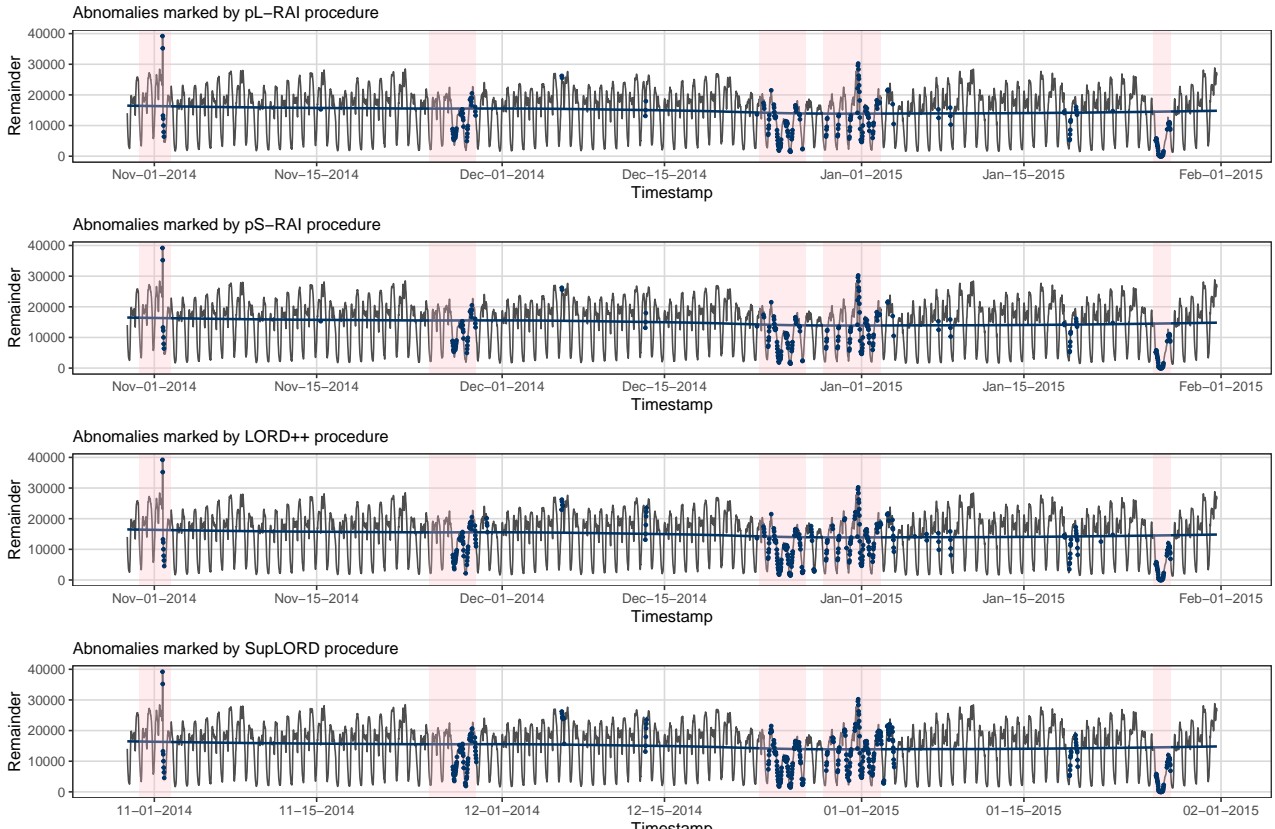

*Figure 11.* Anomaly points detected by pL-RAI, pS-RAI, LORD++, and SupLORD. Rejection points of all procedures are marked by dark blue points. Red regions refer to known anomalies. The testing level is chosen as 0.2.

