# OpenReview forum: "e-GAI: e-value-based Generalized $\alpha$-Investing for Online False Discovery Rate Control"
_ICML.cc/2025/Conference — ICML 2025 poster_

### Official Review · Reviewer_KFoC · 2025-03-11

**Overall Recommendation:** 3

**Summary:**

The paper proposed the e-GAI framework which can control the FDR under arbitrary dependence structures by defining a conservative e-value-based FDP estimator and adopting a risk-averse strategy.

**Claims And Evidence:**

yes

**Essential References Not Discussed:**

no

**Experimental Designs Or Analyses:**

1. In financial bubble detection, ω1 is set to 0.0001 without justification.
2. The paper does not evaluate runtime or memory consumption for ultra-long sequences. I noticed that recursive updates in Eq 8 (where αt depends on all historical Rj ) may lead to linear computational complexity, which might be inefficient in real-world scenarios.

**Methods And Evaluation Criteria:**

yes

**Other Comments Or Suggestions:**

Please see above comments.

**Other Strengths And Weaknesses:**

Strength: Unlike some traditional methods, the proposed e-GAI framework does not need prior knowledge of dependency patterns, which could be more flexible in real-world scenarios.

Weaknesses: 1. The user is required to set the initial parameters (e.g., ω₁, λ), which may rely on experience or a large number of experiments in practical applications.
2. The long-term performance of the algorithm has not been analyzed. For example, if the dynamic allocation strategy (Eq 9) consume too much α-wealth with high rejection rates in the early stages, will the late test sacrifice overall efficacy due to lack of budget, and does it have cumulative errors, etc.?

**Questions For Authors:**

Q1. The covariance matrix assumes equal correlation coefficients (ρ) across all time points, what if there are some time-varying correlations (e.g., increasing ρ) or nonlinear dependencies?
Q2: How to consider the long-term performance of the algorithm?

**Relation To Broader Scientific Literature:**

The paper’s contributions build upon and extend prior work in online false discovery rate control and e-value theory.

**Theoretical Claims:**

no

---

> ### Author Rebuttal · Authors · 2025-04-01
>
> Thank you for your helpful comments and suggestions. We would like to part-wisely respond to your comments.
>
> > In financial bubble detection, ω1 is set to 0.0001 without justification.
>
> This application can be seen as a long-term time series (i.e., $T=10000$). As response to your W1&W2, we suggest choosing $\omega_1$ as $1/T$ approximately to avoid spending too much wealth in the early stages and resulting in $\alpha$-death.
>
> > The paper does not evaluate runtime or memory consumption...
>
> We would like to make the following clarification.
>
> + The update of $\omega_t$ and $\alpha_t$ can be expressed in a recursive form, so computation is highly efficient and memory-friendly. Specifically, updating $\omega_t$ is shown in (9) (line 195). To compute $\alpha_{t}$ for e-LORD (lines 165-177), we define remaining wealth $r_t^{\text{e-LORD}}=\alpha-\sum_{j=1}^{t}\frac{\alpha_j}{R_{j-1}+1}$ and update $\alpha_t=\omega_t r_t^{\operatorname{e-LORD}}(R_{t-1}+1)$. Similar recursive $\alpha_t$ for e-SAFFRON can be obtained.
> + We also evaluated runtimes of various algorithms in Table 2 below. It shows that e-LORD and e-SAFFRON are computationally fast. We will add the recursive update form and present results in final version.
>
> > W1: The user is required to set the initial parameters (e.g., ω₁, λ)...
>
> Thank you for your insightful comments! We would provide some suggestions about $\omega_1$ and $\lambda$.
>
> + In general, a larger $\omega_1$ means more wealth will be assigned to each hypothesis testing, making it easier to discover alternatives, and meanwhile, it will be more possible to exhaust entile wealth at an early stage. Hence, we recommend a relatively small $\omega_1$ for a long period, e.g., $\omega_1\approx 1/T$ empirically.
> + $\lambda$ affects initial wealth and the proportion of hypotheses counted into FDP's estimator. We recommend $\lambda=0.1$ to preserve wealth as discussed in _Remark 3.5_.
> + We have simulation results with different ($\omega_1$,$\lambda$) in Appendix C.1. Additional results of powers of e-LORD and e-SAFFRON ($\lambda>0$) under an $\operatorname{AR}(1)$ model are shown in Table 1 below.  All FDRs are controlled and omitted due to the space limit. We can see the choices of  ($\omega_1,\lambda$) do not affect results too much. So, it enables users to integrate domain knowledge in real applications.
>
> Table 1: Power (%) with different ($\omega_1,\lambda$) and $\alpha=5$%.
>
> |  | e-LORD | $\lambda=0.1$ | $\lambda=0.2$ | $\lambda=0.3$ | $\lambda=0.4$ | $\lambda=0.5$ |
> | --- | --- | --- | --- | --- | --- | --- |
> | $\omega_1=0.001$ | 69.92 | 70.53 | 69.36 | 68.04 | 66.46 | 64.56 |
> | $\omega_1=0.005$ | 67.16 | 75.03 | 74.19 | 72.99 | 71.53 | 69.79 |
> | $\omega_1=0.01$ | 49.41 | 66.47 | 65.69 | 64.54 | 63.19 | 61.48 |
>
> > W2: Long-term performance of algorithm has not been analyzed. ...
>
> As noted, excessive early-stage wealth consumption leads to "$\alpha$-death", halting rejections once $\alpha$-wealth is zero. It is a common phenomenon within GAI framework. So, we recommend using a small $\omega_1$ to preserve wealth as response to your W1.
>
> This issue warrants further research, such as considering "activation" after reaching a certain condition or exploring scenarios where online sequence inherently exhibits block structure, among other possibilities. In addition, this work focuses on measuring the cumulative error by FDR. Exploring other metrics is also a valuable future direction.
>
> > Q1: ...what if there are some time-varying correlations...
>
> We design a new time-varying  $\text{AR}(1)$ model: $X_t=\rho_t X_{t-1}+\epsilon_t$ for $H_0$ and $X_t=4+\rho_t X_{t-1}+\epsilon_t$ for $H_1$ with $\rho_t=\frac{2}{1+\operatorname{exp}(-0.01(t-T/2))}-1$. The table below shows that e-LORD and e-SAFFRON control FDR while leading to relatively high power.
>
> Table 2: FDR, power, and runtime with $\alpha=1$%. SupLORD is referenced by **Reviewer k3m9**.
>
> | | e-LORD | e-SAFFRON | e-LOND | LORD | SAFFRON | SupLORD |
> | --- | --- | --- | --- | --- | --- | --- |
> | FDR (%) | 0.01 | 0.03 | 0.00 | 0.15 | 1.07 | 0.00 |
> | Power (%) | 54.29 | 58.03 | 15.42 | 78.05 | 91.06 | 10.22 |
> | Runtime ($\times 10^{-4}$s) | 11.7 | 17.8 | 17.0 | 11.0 | 5.0 | 77.6 |
>
> > Q2: How to consider long-term performance?
>
> Here we focus on controlling online FDR, a common criterion in online multiple testing. To alleviate "$\alpha$-death" over a long period (potentially infinite), Ramdas et al. (2017) discussed controlling decaying memory FDR under independence. But, how to control it under dependence remains challenging due to additional complexity introduced by these dependencies. As response to your W1, another potential way is to consider other strategies or specific structures. We will include a careful discussion of long-term issues in a future revision per your suggestions. Thank you!

---

> > ### Comment · Reviewer_KFoC · 2025-04-02
> >
> > Thanks for the detailed responses and your new simulation results and clarifications.
> >
> > 1. While the authors recommend setting $ω_1 \approx 1/T$, I could not find any theoretical justification for this choice in the paper. If such justification exists, I would appreciate it if the authors could point it out. From my perspective, the proposed update strategy appears to be heuristic. Moreover, It seems no analysis is provided regarding how the choice of $ω_1$ interacts with $T$, nor are there experimental results across varying T to assess robustness.
> >
> > 2. Regarding Q2 (long-term performance and $\alpha$-death), the authors acknowledge the issue but do not provide any experimental evidence/figures (e.g., α-wealth over time), or even suggested strategies/analysis to address the problem. While referencing Ramdas et al. (2017) for the independence case is useful background, the paper’s setting includes general dependence, where the issue remains unaddressed. I look forward to seeing this discussion later.

---

> > > ### Author Response · Authors · 2025-04-07
> > >
> > > Thank you for your helpful suggestions! We would like to part-wisely respond to your comments.
> > > > Q1: choices of $\omega_1$...
> > >
> > > + In e-GAI, we update $\omega_t$ from a **risk aversion** perspective in (9), _dynamically_ allocating the testing levels and enabling _data-driven_ updates to achieve higher power; see lines 188-193 of right column. In contrast, $\alpha_t$ in e-LOND is derived from a _pre-specified_ decay sequence that sums to 1.
> > > + A simplified version is to set $\omega_t=\omega_1$ for all $t$, and a natural choice is $\omega_1=1/T$, motivating the choice of initial value for our dynamic updates.
> > > + Empirical results support this analysis. Table 1 shows power for different $\omega_1$ across $T$ under the same AR(1) model in previous response. It shows that e-GAI with $\omega_1=1/T$ achieves the highest power, latest last rejection, and largest tail testing level, supporting long-term testing. Moreover, its updates are data-driven: remaining wealth remains robust as $T$ varies. In contrast, $\alpha_T$ of e-LOND diminishes as $T$ increases. Table 2 shows discoveries in real data example in Section 5.2. From Table 2, we observe that allocating an excessively small or large initial wealth results in few rejections.
> > >
> > > Table 1: Results under AR(1) model with $\alpha=0.05$.
> > >
> > > ||Method|$\omega_1$|Power (%)|Time of the last rejection|$\alpha_T/\alpha(\times 10^{-4})$|
> > > |-|-|-|-|-|-|
> > > |$T=500$|e-LORD|$1/T$|70.0|498|1031.0|
> > > |||$1/\sqrt{T}$|22.2|275|0.0|
> > > |||$1/T^2$|8.6|483|0.7|
> > > ||e-SAFFRON|$1/T$|70.5|498|1368.1|
> > > |||$1/\sqrt{T}$|38.7|441|0.0|
> > > |||$1/T^2$|8.0|483|0.6|
> > > ||e-LOND|-|30.9|491|2.5|
> > > |$T=1000$|e-LORD|$1/T$|70.1|998|1029.0|
> > > |||$1/\sqrt{T}$|16.2|406|0.0|
> > > |||$1/T^2$|4.5|962|0.2|
> > > ||e-SAFFRON|$1/T$|70.9|998|1366.7|
> > > |||$1/\sqrt{T}$|28.0|706|0.0|
> > > |||$1/T^2$|4.2|958|0.2|
> > > ||e-LOND|-|23.9|983|1.0|
> > >
> > > Table 2: Number of discoveries with different $\omega_1 (T=8320)$.
> > >
> > > |$\omega_1$|e-LORD|e-SAFFRON|e-LOND|
> > > |-|-|-|-|
> > > |$10^{-4}(O(1/T))$|47|46|33|
> > > |$10^{-8}(O(1/T^2))$|33|33||
> > > |$10^{-2}(O(1/\sqrt{T}))$|3|3||
> > >
> > > > Q2: long-term performance...
> > >
> > > + The e-LORD and e-SAFFRON algorithms dynamically adjust the ratio to allocate _remaining_ wealth (of a fixed budget $\alpha$), thus theoretically leading to the $\alpha$-death when the sequence is infinite. We added experiments to investigate the long-term performance of e-GAI algorithms; see Table 1 in the above response for more settings. Table 3 shows e-GAI outperforms e-LOND in wealth retention and sustained rejections. $\alpha$-death occurs slowly (not yet observed), likely benefitting from data-driven $\omega_t$ update.
> > >
> > > Table 3: Results under AR(1) model with $\alpha=0.05$ and $\omega_1=1/T$.
> > >
> > > ||Method|Power (%)|Time of the last rejection|$\alpha_T/\alpha(\times 10^{-4})$|
> > > |-|-|-|-|-|
> > > |$T=10000$|e-LORD|69.5|9998|1021.1|
> > > ||e-SAFFRON|70.1|9998|1344.6|
> > > ||e-LOND|7.4|9898|0.0|
> > >
> > > + To alleviate $\alpha$-death over a long period, Ramdas et al. (2017) defined decaying memory FDR (mem-FDR) by introducing a user-defined discount factor $d\in (0,1] $ and proposed mem-LORD++ that controls mem-FDR instead under independence.
> > > + To address the issue, we design **mem-e-GAI** to control mem-FDR in our setting as follows. The technique used here is similar to e-GAI framework in the main text. Denote the denominator of mem-FDP as $R_t^m$ for simplicity. A natural choice that is $(j-1)$-measurable for _predicting_ $R_t^m$  is $d^{t-j}[dR_{j-1}^m+1]$. It holds $d^{t-j}[dR_{j-1}^m+1]\leq (R_t^m\vee1)$ for each $\delta_j=1$. Therefore, we propose an **oracle estimate of mem-FDP** as $\sum_{j\in H_0(t)}\frac{\alpha_j}{dR_{j-1}^m+1}$ and design mem-e-LORD by overestimators $\sum_{j=1}^t\frac{\alpha_j}{dR_{j-1}^m+1}$. Thus the testing levels of mem-e-LORD are $\alpha_t=\omega_t(\alpha-\sum_{j=1}^t\frac{\alpha_j}{dR_{j-1}^m+1})(dR_{j-1}^m+1)$ with updating $\omega_t$ in (9) in the main text. Similar results for mem-e-SAFFRON can be obtained, and we omit them due to space limitation. Using the same proof technique as in Appendix A, it can be shown that **both mem-e-LORD and mem-e-SAFFRON achieve mem-FDR control**.
> > > + Table 4 presents the revelant results under AR(1) model. **All mem-FDRs are controlled** and omitted due to the space limit. The mem-Power is the decaying memory power in (Ramdas et al., 2017). From Table 4, mem-e-GAI performs well over long-time testing periods, especially with sparse alternatives, outperforming mem-LORD++ in mem-Power.
> > >
> > > Table 4:  Results under AR(1) model with $\alpha=0.05,d=0.99$, and $\omega_1=1/T$.
> > >
> > > ||Proportion of alternatives $\pi_1$|Method|mem-Power (%)|Time of the last rejection| $\alpha_T/\alpha(\times 10^{-4})$|
> > > |-|-|-|-|-|-|
> > > |$T=10000$|0.01|mem-e-LORD|10.3|8939|0.4|
> > > |||mem-e-SAFFRON|10.3|8939|0.3|
> > > |||mem-LORD++|0.0|1171|0.0|
> > > ||0.05|mem-e-LORD|11.0|9787|0.6|
> > > |||mem-e-SAFFRON|10.6|9782|0.5|
> > > |||mem-LORD++|56.0|9650|55.7|
> > >
> > > We greatly appreciate your comments, which have significantly improved our paper, and we will incorporate them into the revision.

---

### Official Review · Reviewer_k3m9 · 2025-03-13

**Overall Recommendation:** 2

**Summary:**

The paper proposes a framework for online multiple testing with false discovery rate (FDR) control that utilizes e-values with generalized alpha-investing methods to improve power for e-values that satisfy conditional validity. This paper then establishes connections between these methods and existing generalized alpha-investing methods based on p-values, and provide numerical simulations and real data experiments demonstrating the performance of their new methods.

**Claims And Evidence:**

> "The proposed e-GAI can ensure provable online FDR control under arbitrary dependence while improving the power by dynamically allocating the testing levels." (in abstract)

This claim (and variants) are the central thrust of the paper and repeated throughout, though it is incorrect. All FDR controlling results require condition (2), which enforce that the e-values are conditionally valid on past rejection decisions. This is a special case of the conditional superuniformity assumption for p-values that has been repeated throughout the existing literature, and indeed the authors themselves acknowledge it in Appendix B. Prior literature in both online and offline multiple testing have consistently referred to the term "arbitrary dependence" as allowing any kind of dependence structure among p-values/e-values (test statistics) for each hypothesis. Notably, positive dependence among all test statistics (e.g., jointly Gaussian with only positive correlations) *do not* satisfy conditional validity, but fall under the umbrella of arbitrary dependence. The inaccuracy of this claim becomes relevant in the setup choice in numerical simulations (see later point in "Experimental Design and Analyses")

> Theorem 4.1 + "applying LORD++ to $\\{1/e_t\\}$ is equivalent to applying e-LORD to $\\{e_t\\}$" in Appendix B.

I don't believe LORD++ and e-LORD have any equivalence (nor SAFFRON and e-SAFFRON). The $FDP_e^*(t)$ estimator introduced in this paper is much more conservative than the typical $\widehat{FDP}^{\text{LORD}}(t)$ from prior work, since the denominator is smaller in the former. This smaller denominator is a key part of the proof. This discrepancy is noted in the intro by the authors, so it seems that the intro and Appendix B are incoherent. Further, in Theorem 4.1 they require $\mathbb{E}[FDP_e^*(t)] = \mathbb{E}[\sum_{j = 1}^t \alpha_j / (R_t \vee 1)]$, which also is incorrect, since $\mathbb{E}[FDP_e^*(t)]$ is conservative (though this seems like a typo).

**Essential References Not Discussed:**

- Aaron Fisher. Online false discovery rate control for LORD++ and SAFFRON under positive, local dependence. *The Biometrical Journal*, 2024.

This paper shows that LORD++ and SAFFRON have valid FDR control under conditional superuniformity (a more general condition that is implied by the conditional validity in the paper), and an additional positive dependence assumption on the test statistics. This should at least be cited in the context of existing work with FDR control under conditional superuniformity.

- Ziyu Xu and Aaditya Ramdas. Dynamic Algorithms for Online Multiple Testing. *Mathematical and Scientific Machine Learning*, 2022.

The SupLORD algorithm in this paper (to my knowledge) is the only algorithm that has valid FDR control under (just) conditional superuniformity (apart from algorithms valid under arbitrary dependence) and should be compared to in experiments.

**Experimental Designs Or Analyses:**

- As referred to before, the design of the numerical simulation is unclear --- is the covariance positive for all alternatives or all hypotheses? If it is for all hypotheses (as suggested by "The $\Sigma > 0$ is with all diagonal elements being 1 and off-diagonal elements being \rho$"), then the data simulation is does not provide e-values that satisfy condition (2) that is necessary for all proofs of FDR validity for all methods presented in the paper. This setting is then a bit odd for a paper which focuses on methods with provable control of the FDR. Could the authors confirm whether the covariance is positive for just the alternatives (and independent for the nulls) or across all hypotheses?

- The real data experiments are lacking critical details --- in particular, what are the assumptions on the data generating process under the null in each setting, and how are the e-values/p-values explicitly constructed? This is particularly relevant for Section 5.3 since the choice of p-value and e-value seem to be quite different, and it's unclear whether the gap can be explained by that difference.

**Methods And Evaluation Criteria:**

The methods and evaluation criteria (FDR + power) make sense.

**Other Comments Or Suggestions:**

Looking at the proof in Appendix A.1, is there anything in particular that is specific to e-values? It seems like it would hold for p-values that satisfy conditional superuniformity --- that would be a solid result if it were the case.

**Other Strengths And Weaknesses:**

The real data experiments in this paper are seem quite comprehensive, and the authors did a great job presenting visualizations for the analysis.

**Questions For Authors:**

No questions/concerns outside ones stated. The paper generally seems unclear about what its actual contributions are, and missing comparisons to key prior work/comparisons.

**Relation To Broader Scientific Literature:**

Although the presentation of the paper has somewhat obfuscated what the contributions are, I believe there is an interesting contribution here where the authors have designed a new FDP estimator that allows for online multiple testing with FDR control under conditional validity, at the cost of some power compared to when independence is assumed. This then allows them to use generalized alpha-investing to design procedures valid in that specific setting.

**Theoretical Claims:**

The errors with the theoretical claims are discussed in the prior section.

---

> ### Author Rebuttal · Authors · 2025-04-01
>
> We appreciate your careful reading and constructive suggestions. Per your comments, we would like to clarify key **contributions** of our work as follows:
>
> 1. We **propose the e-GAI framework** for online testing with using e-values, which **achieves FDR control under arbitrary dependence among valid e-values** (i.e., satisfying condition (2)).
> 2. We designed **a new e-value-based FDP estimator** and **provided theoretical guarantees for online FDR control** building on it under the relevant conditions.
> 3. Within the e-GAI framework, we **developed a new updating approach for allocating testing levels from a risk aversion perspective,** aiming to save budget while achieving high power. Along this line, we proposed two new algorithms (**e-LORD and e-SAFFRON**) for implementation.
>
> Thanks for your comments on our paper. We would like to part-wisely respond to your comments.
>
> > This claim (and variants) are central thrust of the paper and repeated throughout...
>
> We sincerely apologize for any confusion. As you have pointed out, by "arbitrary dependence", we mean allowing any dependence structure among test statistics for each hypothesis, i.e., online conditionally valid p-values/e-values.  In the revision, we will refine descriptions to minimize confusion. Further details on the experimental design will be clarified in the following. Thank you!
>
> > I don't believe LORD++ and e-LORD have any equivalence (nor SAFFRON and e-SAFFRON). ...
>
> Thank you for your insightful comments. We agree with you that e-GAI is a different framework from GAI. The "equivalence" in Section 4 was to explore the connection between LORD++ and e-LORD when _independent_ p-values are available. In such cases, we can design testing algorithms using a less conservative FDP estimator, i.e., $\sum_{j = 1}^t \alpha_j / (R_t \vee 1)$ such as the one in Theorem 4.1, which will coincide LORD++ and e-LORD.  Previously, for notational simplicity, Theorem 4.1 did not explicitly highlight the change in the FDP estimator. The final version will revise the discussion in Section 4 for clarity and precision. Hope these interpretations can ease your doubts!
>
> > As referred to before, the design of the numerical simulation is unclear. ...
>
> Thank you for pointing this out! We apologize for the unclear setup description. The covariance matrix assumes positive correlations between all elements, including nulls and alternatives. In simulations, we assessed our methods using oracle p-values/e-values, calculated from the known conditional distributions derived from the normal distribution. Furthermore, we conducted additional experiments by designing other dependence structures, and these results will be added into the future revision. For details, please see our response to **Reviewer kFoC**'s Q1 due to the space limitation.
>
> > The real data experiments are lacking critical details. ...
>
> We appreciate your feedback and offer the following clarification. In Section 5.2, the data is assumed to be independent and normally distributed under $H_0$ as discussed in (Ahmad et al., 2017), using p-values derived from the normal distribution. We employed KDE to estimate $f_t(x)$ and $f_0(x)$, taking their ratio as the e-value. Due to the independence assumption, these e-values are valid.
>
> In Section 5.3, following [1] modeling framework for financial bubbles, we characterized the problem as a right-tailed unit root test in an $\text{AR}(1)$ model. We built valid e-values on normalized likelihood ratios by dividing its conditional expectation under $H_0$.  Both this likelihood ratio and p-values are proposed by Dickey & Fuller's series works. We will clarify these details in the future revision.
>
> > Two References Not Discussed
>
> Thank you for your recommendation! We have introduced and cited Fisher (2024) in Section 2.2 (lines 126-129 of right column) in the main text. Additionally, we include a comparison with the SupLORD algorithm (Xu & Ramdas, 2022) in Table 2 of our response to **Reviewer kFoC**'s Q1 due to space constraints. In the new setting, SupLORD controls FDR but has low power, whereas our methods achieve FDR control with higher power. In future revisions, we will incorporate Xu & Ramdas (2022) and add comparative experiments.
>
> > Looking at the proof in Appendix A.1, is there anything in particular that is specific to e-values? ...
>
> The proof in Appendix A.1 does indeed rely on a unique property of e-values: the larger the e-value, the more significant the evidence against the null. Note that $\delta_t=\mathbb{I}[e_t\geq 1/\alpha_t]\leq e_t\alpha_t$ deterministicly in the numerator of inequality (ii) of the proof. This, however, does _not hold_ for p-values. Hence, e-GAI cannot theoretically guarantee FDR control when directly applied to p-values satisfying condition (1).
>
> Ref:
>
> [1] Phillips, P. C., & Yu, J. Dating the timeline of financial bubbles during the subprime crisis. Quantitative Economics, 2(3), 455-491, 2011.

---

> > ### Comment · Reviewer_k3m9 · 2025-04-05
> >
> > > We propose the e-GAI framework for online testing with using e-values, which achieves FDR control under arbitrary dependence among valid e-values (i.e., satisfying condition (2)).
> >
> > I'd like to re-iterate that *arbitrary dependence* is explicitly used to describe unknown dependence, and condition (2) has been explicitly characterized as *conditional superuniformity*. Conditional superuniformity is a well-studied notion in online multiple testing --- indeed this condition already appears in the origins of the problem in the foundational papers of Foster and Stine (2007) and Saharoni and Rosset (2014). Both papers showed that alpha-investing and GAI provide valid online marginal FDR control at stopping times (a stronger guarantee for the marginal variant of FDR). A characterization that is more accurate may be to say that a known dependence/distribution structure implies conditional superuniformity (vs. online multiple testing methods that control FDR even when the FDR is unknown). This is not simply a semantic difference --- even when the dependence is known, the online multiple testing methods that have validity under unknown arbitrary dependence (reshaped LOND of Zrnic et al. (2021) and e-LOND) also control FDR at stopping times (and not just fixed times) [1].
> >
> > I apologize if the above comment was redundant wrt my original review, but I think method comparison in Table 1 somewhat omits key details about the state of existing work, and consequently the description of condition (2) as "arbitrary dependence" muddies the water wrt the precise contribution made by this work.
> >
> > > This, however, does not hold for p-values.
> >
> > I don't think that part of the proof is critical --- you can accomplish the same thing w/ conditionally superuniform p-values
> >
> > $$\mathbb{E}[\mathbf{1}\\{p_j \leq \alpha_j\\}/ (R_{j - 1} + 1)] = \mathbb{E}[\mathbb{E}[\mathbf{1}\\{p_j \leq \alpha_j\\} \mid \mathcal{F}\_{j - 1}] / (R_{j - 1} + 1)]\leq \mathbb{E}[\alpha_j / (R_{j - 1} + 1]$$
> >
> > where the 2nd step + last step is by conditional superuniformity of $p_j$ and the fact that $\alpha_j, R_{j - 1}$ are measurable wrt $\mathcal{F}_{j - 1}$.
> >
> > [1]  Lasse Fischer and Aaditya Ramdas. An online generalization of the e-BH procedure. arXiv:2407.20683, 2024.

---

> > > ### Author Response · Authors · 2025-04-07
> > >
> > > Thank you for your helpful comments and suggestions. We would like to part-wisely respond to your comments.
> > >
> > > > Q1: I'd like to re-iterate that arbitrary dependence is explicitly used to describe unknown dependence, and condition (2) has been explicitly characterized as conditional superuniformity. ...
> > >
> > > Thanks for your comments. We sincerely apologize for the confusion caused by the ambiguous expression here. In the revised version, we will modify the description to accurately state that our e-GAI algorithm achieves FDR control under conditional superuniformity, revise the comparison of algorithms in Table 1 in the main text, and supplement the relevant and necessary references mentioned in your comment.
> > >
> > > > Q2: I don't think that part of the proof is critical. ...
> > >
> > > We sincerely apologize for overlooking this property and greatly appreciate your insightful observation. The existence of this property will enrich and elevate the contributions and applicability of our methodology framework, making the theory more comprehensive and complete. We would like to add a discussion of this part in the revised version. Thank you very much for your suggestion!

---

### Official Review · Reviewer_PQaa · 2025-03-14

**Overall Recommendation:** 4

**Summary:**

This paper proposes a framework for generalized $\alpha$-investing (GAI) with e-values, an approach for online multiple hypothesis testing.  While prior work had considered GAI with p-values, this paper contributes two things:  First (Section 3.1) they derive bounds on the false discovery proportion (FDP) for the setting of arbitrarily dependent e-values, and second (Section 3.2) they design a schedule for "spending" alpha that depends on previous costs.  While the theory speaks to the false positive rate control, experiments on simulated and real-world data demonstrate that the proposed schedule yields greater power than existing approaches.  There are several baselines considered:  SAFFRON and LORD++ are both based on an assumption of independent p-values (or p-values satisfying the PRDS condition), and so do not have guarantees in the setting of arbitrary dependence, while e-LOND does allow for arbitrary dependence, but uses a spending schedule that does not incorporate prior costs, and empirically has lower power as a result.

**Post rebuttal update**:  See my comments in the chain below.
> After reading the response and the other reviews, I will maintain my score, though I have somewhat lower confidence in my assessment after reading the review of k3m9. Nonetheless, I'm still in favor of the paper being accepted, assuming that these clarifications (and those made to other reviewers) are incorporated.

**Claims And Evidence:**

Yes, the claims are supported by clear and convincing evidence.

**Essential References Not Discussed:**

I did not notice any essential references not discussed, though I am less familiar with this literature, so I may be missing something.

**Experimental Designs Or Analyses:**

Yes, I found both the synthetic (Section 5.1) and the real-data experiments (Sections 5.2, 5.3) to be sound and valid.  Both experiments demonstrate control of the real (in 5.1) and estimated (in 5.2,5.3) false discovery rates based on known anomalies.  I found both real-world experiments easy to follow and compelling.

**Methods And Evaluation Criteria:**

Yes, the evaluation criteria make sense to me.  The theoretical claims demonstrate control over the FDR, and experiments are used to assess power in real-world situations, both of which I would expect to see in a paper like this one.

**Other Comments Or Suggestions:**

It may be worth taking an editing pass for grammar, e.g., Line 092, left-hand column "though this operation only achieves a smaller improvement in power while introduces additional" sounds off, should perhaps be something like "though this operation only achieves...while **introducing** additional" or "though this operation only achieves a smaller improvement in power, **and introduces** additional"

**Other Strengths And Weaknesses:**

First, the empirical argument would have been stronger in Section 5.1 with a simulated scenario that caused LORD++ to exceed the desired FDR bounds.

Second, these are all very minor, but my main critiques of this paper would be related to clarity of presentation, with a few examples where more context would have been helpful:
* Section 2.2, it is stated that the "key idea in GAI...is that each rejection gains some extra $\alpha$-wealth", shouldn't this be the opposite, that accepting the null gains wealth, and rejections spend it?
* I get the math, but had trouble following the conceptual explanation below Theorem 3.1, that $R_{j-1} + 1$ "predicts" the number of possible future rejections, since it obviously under-predicts the total number of rejections. As I understand it, the key is to observe that the "right" denominator is something like $R_t \lor 1$, and that by under-predicting this quantity, we get an upper bound (since we're in the denominator).  A little more explanation would have been helpful here.
* The PRDS condition is mentioned in several places in the introduction / Section 2 (see page 1, "p-values with...positive regression dependence on a subset") but is not formally defined, might have been nice to at least include in the appendix for completeness.

**Questions For Authors:**

There is one area where I wasn't as clear on the contribution:  Could the authors clarify how their results in Theorem 3.1 differ from those in the e-LOND paper?  I didn't see this exact result anywhere in that paper on a very brief skim, but I imagine they need to have used some similar result to also achieve FDR control with arbitrary dependence?

**Relation To Broader Scientific Literature:**

The key contributions appear a little incremental to me, but are summarized well in introduction (Table 1 and "Related Works").  Both prior approaches to GAI with p-values do not allow for arbitrary dependence between p-values, and the existing approach with e-values (e-LOND) does not consider scheduling based on prior costs, and empirically (as shown in experiments) has lower power as a result.

**Theoretical Claims:**

I checked the proofs in Appendix A for the main results, which appear correct to me.

---

> ### Author Rebuttal · Authors · 2025-04-01
>
> Thanks for your comments on our paper. We would like to part-wisely respond to your comments.
>
> > First, the empirical argument would have been stronger in Section 5.1 with a simulated scenario that caused LORD++ to exceed the desired FDR bounds.
>
> Thank you for your valuable suggestion! We have to acknowledge that we explored various simulation settings and did not get the results that LORD++ exceeds the target FDR level for now. This is mainly because the LORD++ algorithm itself is inherently conservative. However, we found that LORD++ does not perform well with $\widehat{\text{FDP}}=0.189$ in the real data analysis in Section 5.2. As we know, LORD++ lacks reliable theoretical guarantees with the dependence of p-values, which may compromise its safety and reliability in practical applications.
>
> > Section 2.2, it is stated that the "key idea in GAI...is that each rejection gains some extra $\alpha$-wealth", shouldn't this be the opposite, that accepting the null gains wealth, and rejections spend it?
>
> We would like to make the following clarifications.
> + The GAI algorithm is designed from an investment perspective: each hypothesis test is treated as an investment, where the significance level $\alpha_t$ is drawn from a wealth pool $W_t$. Thus, each test consumes a certain amount of "wealth", and if the test results in a rejection, it is considered a successful investment, thereby gaining additional "wealth" and increasing the cumulative wealth.
> + In contrast, our e-GAI framework adopts a different design philosophy: e-GAI views the entire testing process as a risky investment and the whole wealth does not increase; instead, each test consumes a fixed proportion of the remaining wealth (e.g., $\omega_t$ in e-LORD). Therefore, each rejection introduces risk, and from a risk-averse perspective, it is necessary to reduce the investment proportion in subsequent tests to ensure long-term sustainability.
> + The difference in design philosophy between e-GAI and GAI stems from the distinct estimators used for the FDP. For the GAI method, which allows for wealth accumulation, increased investment is advantageous. In contrast, for the e-GAI method, where total wealth cannot increase, each investment inherently entails risk.
>
> > I get the math, but had trouble following the conceptual explanation below Theorem 3.1, ... A little more explanation would have been helpful here.
>
> Thank you for your suggestion! As you noted, the key role of $R_{j-1}+1$ here is that it serves as a $(j-1)$-measurable lower bound for $R_t\vee1$. Since $R_{j-1}+1\leq R_t\vee 1$ and it is placed in the denominator, it results in an overestimation of the oracle FDP. As the true $R_t\vee1$ is _unobservable_ at time $j-1$, we use $R_{j-1}+1$ as a substitute, effectively "_predicting_" the value at time $t$ from the perspective of time $j-1$. This is why we used the verb "predict" in this context. Per your comment, we will provide a more detailed explanation in the final version to make this point clear.
>
> > The PRDS condition is mentioned in several places in the introduction / Section 2 (see page 1, "p-values with...positive regression dependence on a subset") but is not formally defined, might have been nice to at least include in the appendix for completeness.
>
> We appreciate your attention to detail and your valuable feedback. We will include a formal definition of PRDS and relevant references in the appendix of the final version.
>
> > It may be worth taking an editing pass for grammar, e.g., Line 092, left-hand column ...
>
> Thank you for pointing this out! We will carefully review the grammar throughout the paper and make the necessary corrections, including the sentence you highlighted.
>
> > Could the authors clarify how their results in Theorem 3.1 differ from those in the e-LOND paper? I didn't see this exact result anywhere in that paper on a very brief skim, but I imagine they need to have used some similar result to also achieve FDR control with arbitrary dependence?
>
> Theorem 3.1 introduces an estimator for the FDP and proves that FDR control can be achieved when the expectation of this estimator does not exceed $\alpha$. Based on this key estimator, we subsequently propose algorithms for designing testing levels (e-LORD and e-SAFFRON). In contrast, the e-LOND algorithm follows a different approach. Specifically, e-LOND does not introduce an FDP estimator; instead, inspired by the LOND algorithm [1], it defines the testing levels $\alpha_t$ by directly leveraging a decay sequence that sums to 1 and the number of rejections. This approach allows for the cancellation of a common term in the numerator and denominator of the FDP, thereby achieving FDR control.
>
> Ref:
> [1] Zrnic, T., Ramdas, A., and Jordan, M. I. Asynchronous online testing of multiple hypotheses. Journal of Machine Learning Research, 22(33), 1-39, 2021.

---

> > ### Comment · Reviewer_PQaa · 2025-04-07
> >
> > Thank you for these clarifications.  After reading the response and the other reviews, I will maintain my score, though I have somewhat lower confidence in my assessment after reading the review of k3m9.  Nonetheless, I'm still in favor of the paper being accepted, assuming that these clarifications (and those made to other reviewers) are incorporated.

---

> > > ### Author Response · Authors · 2025-04-08
> > >
> > > We sincerely appreciate your helpful comments and continued support for our work. Addressing your thoughtful concerns has helped us improve the presentation of our paper. We will incorporate all clarifications, including responses to all reviewers, into the revised version of the paper to enhance its clarity and quality. Thank you again for your valuable suggestions and your participation in reviewing our work!

---

### Official Review · Reviewer_caTX · 2025-03-16

**Overall Recommendation:** 3

**Summary:**

The paper extends the generalized $\alpha$-investing (GAI) framework for online testing by allowing it to be based on e-values as well (hence the name e-GAI). This allows for online false discovery rate control under arbitrary dependencies of the hypotheses and can lead to improved power under good dynamic allocation of the test levels. More concretely, they do this by defining an oracle estimate of the false discovery proportion at each time that is based on e-values. Then, they employ existing methods of bounding it from the literature, leading to the corresponding methods e-LORD and e-SAFFRON. They supplement their proposal by numerical experiments (including a simulation, taxi anomaly detection, and financial bubble detection).

**Claims And Evidence:**

Yes.

**Essential References Not Discussed:**

Not to my knowledge.

**Experimental Designs Or Analyses:**

I found the experimental analyses to be thorough and well-done. I appreciated the inclusion of different types of data (simulated and real). One small note is that maybe Figure 2 could benefit from some improvements in its display as currently the squares and dots are quite small and subtle (the latter could be fixed by different coloring I think).

**Methods And Evaluation Criteria:**

Yes.

**Other Comments Or Suggestions:**

One general small personal suggestion would be to not center the A/B testing for tech companies applications so much in the abstract and introduction, and instead showcase more exciting (possibly for social good) applications.

Typos:
- add “The” before “e-LOND” in line 19, left column
- add “the" before “GAI” in line 70 right column
- “guarantee” should be “guarantees” in line 417 right hand side

**Other Strengths And Weaknesses:**

Beyond the explicit contributions, an additional strength is that the proposed framework unifies multiple existing approaches. A potential weakness is that the paper doesn't go into too much depth on devising new methods, and mainly instantiates LORD and SAFFRON in their own framework.

**Questions For Authors:**

N/A

**Relation To Broader Scientific Literature:**

This paper is connected to the broader scientific literatures of sequential hypothesis testing via betting and, especially, the line of work in online testing with FDR control. The authors do a good job of discussing and comparing to prior methods.

**Theoretical Claims:**

I skimmed through the proofs and they appear correct.

---

> ### Author Rebuttal · Authors · 2025-04-01
>
> Thanks for your comments on our paper. We would like to part-wisely respond to your comments.
>
> > One small note is that maybe Figure 2 could benefit from some improvements in its display as currently the squares and dots are quite small and subtle (the latter could be fixed by different coloring I think).
>
> Thank you for your advice! We have improved Figure 2 by increasing the size of the squares and dots and adjusting the coloring for better contrast. These changes will present in the future revision.
>
> > A potential weakness is that the paper doesn't go into too much depth on devising new methods, and mainly instantiates LORD and SAFFRON in their own framework.
>
> Thank you for your comments! Our work aims to address the challenge of online FDR control under arbitrary dependence. To overcome the limitations of p-values in this context, we propose a novel framework leveraging e-values for hypothesis testing, along with two concrete and implementable algorithms, i.e., e-LORD and e-SAFFRON. Furthermore, under the independence assumption, we establish connections between our framework and existing methods, such as LORD++ and SAFFRON from the GAI framework. We have named our approach _e-GAI_, which reflects the key method is to build a new GAI framework with the use of e-values under arbitrary dependence.
>
> > One general small personal suggestion would be to not center the A/B testing for tech companies applications so much in the abstract and introduction, and instead showcase more exciting (possibly for social good) applications.
>
> Thank you for this suggestion! We will revise the abstract and introduction to emphasize additional use cases, such as real-time monitoring of machine operation status in industrial settings. Indeed, we have applied the e-GAI algorithm to one relevant real example in Appendix C.2. We appreciate that this will better showcase the potential benefits of our framework.
>
> > Typos: add “The” before “e-LOND” in line 19, left column; add “the" before “GAI” in line 70 right column; “guarantee” should be “guarantees” in line 417 right hand side
>
> Thank you for the careful reading! We will correct these typos in the final version.

---

### Decision · Program_Chairs · 2025-05-01

**Decision:**

Accept (poster)

**Comment:**

This paper studies a framework for online multiple testing with false discovery rate (FDR) control that utilizes e-values and generalized alpha-investing methods to improve power when the e-values satisfy conditional validity. It then establishes connections between these methods and existing generalized alpha-investing approaches based on p-values, and provides numerical simulations and real data experiments demonstrating the performance of the proposed methods.

Reviewer k3m9 points out a few issues in both the writing (e.g., conditional superuniformity) and the experiments (e.g., SupLORD). Upon digging into it, I tend to agree with Reviewer k3m9's comments. To ensure this work be rigorous and sound, I believe the paper needs to be revised before it can be published. The response and constructive feedback (e.g., discussion on parameter choices) from other reviewers should also be considered in the next version.